# Twist piezoelectricity: giant electromechanical coupling in magic-angle twisted bilayer LiNbO$_3$

Hulin Yao [1,2,5], Pengcheng Zheng[1,2,5], Shibin Zhang [1,5] ✉, Chuanjie Hu[3], Xiaoli Fang[1,2], Liping Zhang[1], Dan Ling[1,2], Huanyang Chen [3,4] ✉ & Xin Ou [1,2] ✉

Twisted a pair of stacked two-dimensional materials exhibit many exotic electronic and photonic properties, leading to the emergence of flat-band superconductivity, moiré engineering and topological polaritons. These remarkable discoveries make twistronics the focus point of tremendous interest, but mostly limited to the concept of electrons, phonons or photons. Here, we present twist piezoelectricity as a fascinating paradigm to modulate polarization and electromechanical coupling by twisting precisely the stacked lithium niobate slabs due to the interlayer coupling effect. Particularly, the inversed and twisted bilayer lithium niobate is constructed to overcome the intrinsic mutual limitation of single crystals and giant effective electromechanical coupling coefficient $k_t^2$ is unveiled at magic angle of 111$_o$ reaching 85.5%. Theoretical analysis based on mutual energy integrals shows well agreements with numerical and experimental results. Our work opens new venues to flexibly control multi-physics with magic angle, stimulating progress in wideband acoustic-electric, and acoustic-optic components, which has great potential in wireless communication, timing, sensing, and hybrid integrated photonics.

With the discovery of unconventional flat bands[1,2] induced by the strong correlations in magic-angle bilayer graphene, van der Waals materials stacked with a relative twist angle have become an unprecedented platform in the condensed matter physics community, leading to various complex emergent quantum phenomena such as superconductivity[1,3–5], ferromagnetism[6,7], quantum anomalous Hall effect[8] and correlated insulating phases[2,9]. Analogous concepts have made many novel advances in the field of photonics, by the recent demonstrations of atomical photonic crystals in moiré graphene superlattices[10], transition-metal dichalcogenides[11–13], topological transitions were found in twisted α-phase molybdenum trioxide bilayers[14–16]. The principle behind these exotic phenomena, which is associated with the formation of moiré superlattice in twisted van der

Waals materials, comes from the interlayer coupling of wave responses and varies with the twist angle between bilayers. Magic-angle materials have exhibited the ability to transcend the limitations of nature material crystals, offering unprecedented flexibility in electronic, photonic, and phononic field manipulation.

Piezoelectricity characterized by the piezoelectric constitutive relations is the linear coupling effect of the electromechanic fields in non-centrosymmetric crystals and is determined by lattice symmetry. The conversion of mechanical energy into electrical energy and vice versa, is of extensively used in wireless communications[17–19], acousto-optic modulation[20–22], bioacoustics[23,24], nanoacoustics[25–28], etc. Particularly, RF bandpass filters with larger physical bandwidth are urgently needed to significantly expanded data capacity as the rapid

[1]State Key Laboratory of Materials for Integrated Circuits, Chinese Academy of Sciences, 865 Changning Road, Shanghai 200050, China. [2]The Center of Materials Science and Optoelectronics Engineering, University of Chinese Academy of Sciences, 19 Yuquan Road, Beijing 100049, China. [3]Department of Physics & Department of Microelectronics and Integrated Circuit, Xiamen University, 422 Siming South Road, Xiamen 361005, China. [4]Department of Physics, Xiamen University Malaysia, Sepang, Malaysia. [5]These authors contributed equally: Hulin Yao, Pengcheng Zheng, Shibin Zhang.
✉e-mail: sbzhang@mail.sim.ac.cn; kenyon@xmu.edu.cn; ouxin@mail.sim.ac.cn

development of 5th generation (5 G) and future 6 G wireless communication technology. The efficiency of energy conversion in the piezoelectric materials, measured by the electromechanical coupling coefficient ($k_t^2$) or effective coupling coefficient ($k_{eff}^2$), is positively correlated with the fractional bandwidth of the acoustic wave filters. One of particular interest questions is the selection of piezoelectric crystals and their cuts type to achieve the highest energy conversion efficiency.

Recently, owing to large values of piezoelectric stress constants, flexible electro-acousto-optic modulation characteristics, and extremely low optical transmission loss, lithium niobate (LN) was widely studied and employed in RF signal processing, optics, acoustic-optics, acoustic-quantum, etc. The strong anisotropy of LN allows to tune the $k_{eff}^2$ ($k_t^2$) through choosing an optimized crystal orientation while maintaining high operating frequency and spurious free resonance responses. However, the performance of piezoelectric coupling is limited by the intrinsic crystal constants, and it is highly desirable to break this predicament inspire by the concept of magic angle.

Here we report an observation of piezoelectrical-excited in-plane polarized acoustic modes in inversed and twisted placed bilayer lithium niobate structure (ITBLN), in which the resonance frequencies and coupling coefficients can be adjusted through changing the twisted angle and thickness ratio rather than changing crystal orientations. With electrodes attached on the top and bottom of the bilayer lithium niobate, a longitudinal electric field was applied into the two piezoelectric crystals, and corresponding piezoelectrical acoustic waves were excited and superposed to form a new series of standing resonance modes. As theoretical analyzed, the effective electromechanical

coupling $k_t^2$ of one of the two lowest-order modes would reach its maximum as large as 85.5% at extreme point on dispersion mapping curves while the other diminish to almost zero. Our work provides a new prospect to break the intrinsic limitation of single layer crystal and expanding horizons for acoustic structures with multilayered modulated piezoelectricity.

## Results

### Angular dispersion of the two lowest order modes

Figure 1a shows the schematic structure of the single layer X-cut LN while Fig. 1b shows the proposed ITBLN, which consisting of two X-cut lithium niobate layers with their crystal-axis placed inversely and twistedly. The admittance responses of the single layer LN and the ITBLN with different twisting angles are measured and shown in Fig. 1c. Only the first two fundamental acoustic modes in curves are studied here, since they correspond to two piezoelectrically driven acoustic modes in ITBLN, marked as mode I and mode II as well as $\alpha$ and $\beta$ polarization types in single layer LN. An important distinction confirmed by the curves is that the resonant frequency ($f_r$) and electromechanical coupling coefficients ($k_t^2$) of the two modes can be modulated through adjusting the structure parameters such as the thickness ratio of bilayer layers and the twisting angle. As a comparison, the admittance response of single layer LN exhibits one main peak, corresponding to $\alpha$-polarization mode with the largest $k_t^2$, but exists a remarkable spurious peak owing to $\beta$-polarization mode. This spurious response would dramatically degrade the performance of devices, and moreover, is lack of ability of modulation. This limits the utilization potentiality in wideband operations and novel phonon-electron interactions.

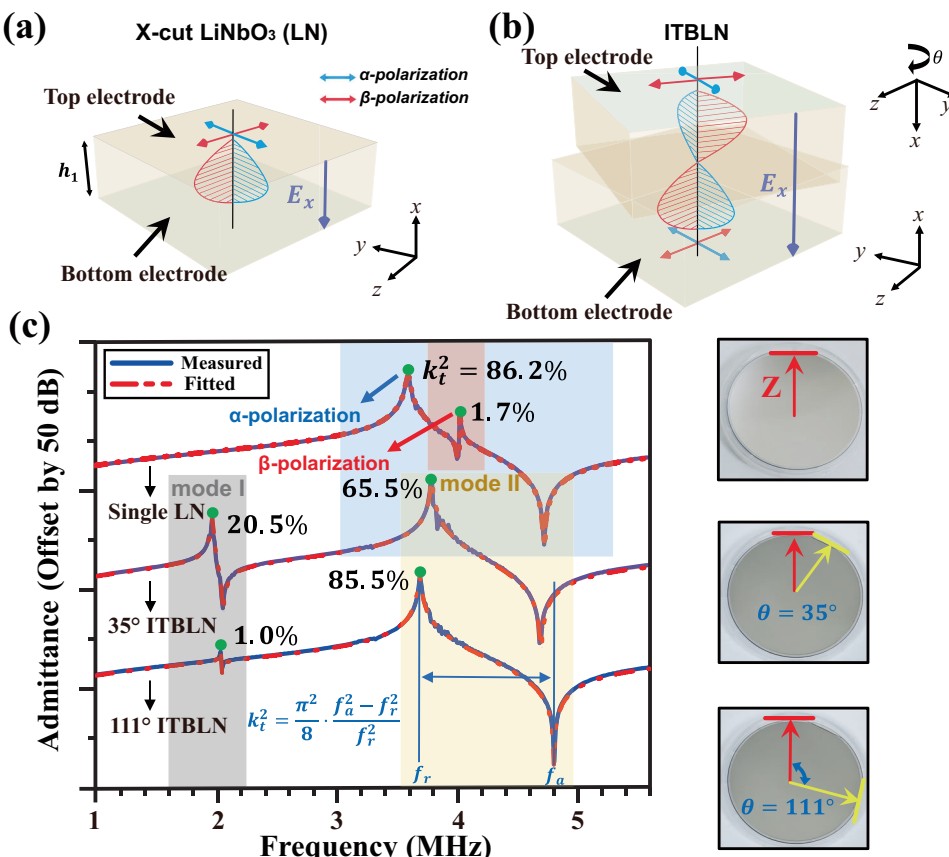

**Fig. 1 | Schematics and electrical responses of piezoelectrically excited acoustic resonant cavities which consisting of single layer X-cut LN and inversed and twisted bilayer lithium niobate layers (ITBLN). a** Schematic of single layer X-cut LN structure, the wave packets consisting of arrows in red and blue color represents two orthogonal-polarized acoustic eigenmodes in LN, with longitudinal electric field piezoelectrically associated and provided by top and bottom electrode. **b** Schematic of ITBLN structure, the crystal cut of bilayer LN was set to make an opposite directing x-axis through inversion with z-axis and twisting. **c** Experimental admittance curves of single layer LN and ITBLN at different twisting angles, the thickness of both LN layers are 500 um, equaling to the total thickness of 1000 um of single layer LN.

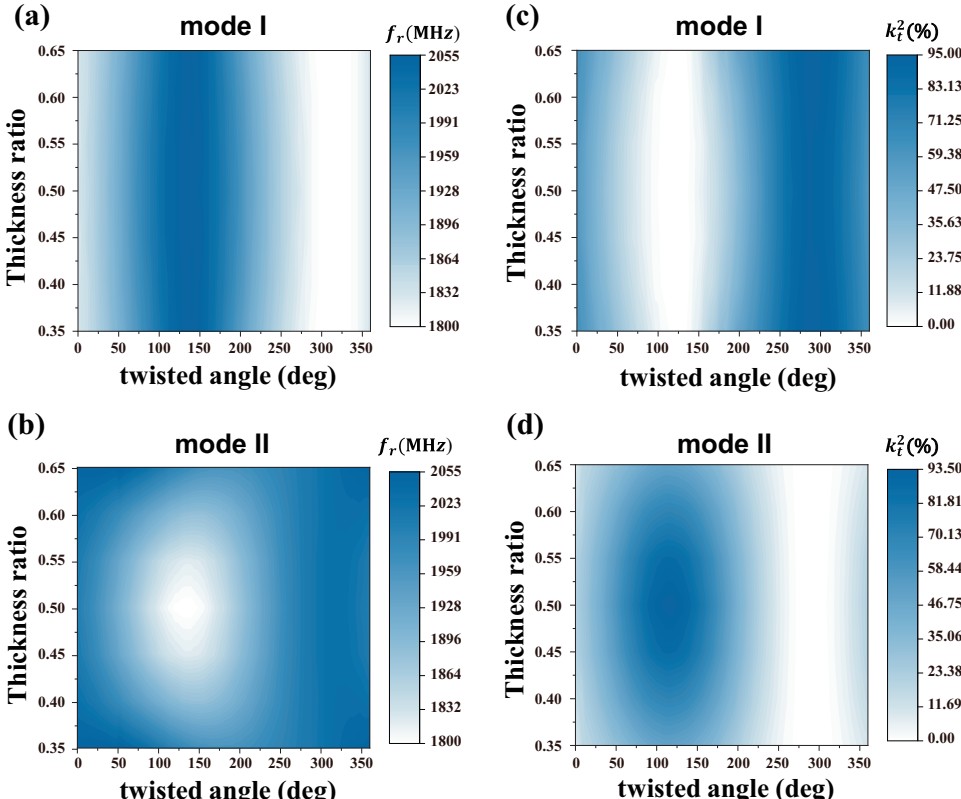

**Fig. 2 | Dispersion and energy conversion efficiency of first two shear-polarized modes via structural parameters.** Simulated resonant frequency of (**a**) mode I and **b** mode II via thickness ratio of two piezoelectric layers and twisted angle. Simulated electromechanical coupling coefficient of (**c**) mode I and **d** mode II via thickness ratio of two piezoelectric layers and twisted angle.

A further investigation of the two modes were studied using periodic finite-element-method based (FEM) simulation model. The simulated resonant frequency $f_r$ and electromechanical coupling coefficient $k_t^2$ maps of the two modes versus the twisted angle and thickness ratio of two piezoelectric layers are shown in Fig. 2, respectively. The dispersion characteristics are significantly different, as mode I maintaining a minor disturbance with changing of thickness ratio, namely a band shaped dispersion map, while mode II exhibiting an extremum point and an oval shaped dispersion map. On the other hand, when the thickness of the two piezoelectric layers is the same, both modes have an optimal angle for maximization of coupling coefficients respectively. The optimal angle was about 296 degrees for mode I and 118 degrees for mode II.

**Theoretical analysis of piezoelectrically coupled waves in twisted bilayer lithium niobate structure**

The piezoelectrically coupled fields equations in anisotropic medium can be deduced from electromagnetic and acoustic fields equations of each own and simultaneously the piezoelectric constitutive relations (see Supplementary Note 1). Taking the displacement fields **u** and the electric fields **E** as fields variables, the coupled wave equations can be expressed as[29]:

$$\nabla \cdot \mathbf{c}^E : \nabla_s \mathbf{u} - \rho \frac{\partial^2 \mathbf{u}}{\partial t^2} = \nabla \cdot \mathbf{e} \cdot \mathbf{E} \tag{1}$$

$$\nabla \times \nabla \times \mathbf{E} + \mu_0 \boldsymbol{\varepsilon}^S \cdot \frac{\partial^2 \mathbf{E}}{\partial t^2} = -\mu_0 \mathbf{e} : \frac{\partial^2}{\partial t^2} \nabla_s \mathbf{u} \tag{2}$$

where $\mathbf{c}^E$ is elastic constants at zero or constant electric field, $\rho$ is density of material, **e** is piezoelectric stress constants, $\mu_0$ is permeability of vacuum, $\boldsymbol{\varepsilon}^S$ is permittivity at zero or constant strain. The operation of ":" in equations is defined as contraction of tensor. Equations (1) and (2) are solved under X-propagating condition for time hormonic waves in Supplementary Note 1. According to the directions of polarization and coupling of different components of electric fields and elastic fields, the solutions contain three acoustic waves: (i) one longitudinal polarized wave coupled with in-plane polarized electromagnetic waves, called as quasi-longitudinal acoustic wave, (ii) and (iii) two orthogonal shear horizontal (SH) polarized waves with different phase velocities, coupled with longitudinal irrotational electric field distributions, called as piezoelectric stiffened SH acoustic waves and were of concerned in this letter. Noted that the wave vector **k** was determinative in X-crystal orientation as $(k,0,0)$ in crystal axis system. As shown in Fig. 3a, the polarization type of quasi-longitudinal wave can be treated as a combination of electromagnetic wave and longitudinal acoustic wave. The phase velocity solved by its dispersion equation gives three values, and one of them is in the typical level of elastic waves, and the other two correspond to the ordinary light and the unusual light in LN crystal. Meanwhile, the phase velocity of the two orthogonal piezoelectric stiffened SH waves are different, but both satisfy the Christoffel-type dispersion equation (See Supplementary Note 1), which can be derived as:

$$k^2 \begin{bmatrix} c_{66} + \frac{e_{16}^2}{\epsilon_{11}} & c_{56} + \frac{e_{15}e_{16}}{\epsilon_{11}} \\ c_{56} + \frac{e_{15}e_{16}}{\epsilon_{11}} & c_{55} + \frac{e_{15}^2}{\epsilon_{11}} \end{bmatrix} \cdot \begin{bmatrix} u_y \\ u_z \end{bmatrix} = \rho \omega^2 \begin{bmatrix} u_y \\ u_z \end{bmatrix} \tag{3}$$

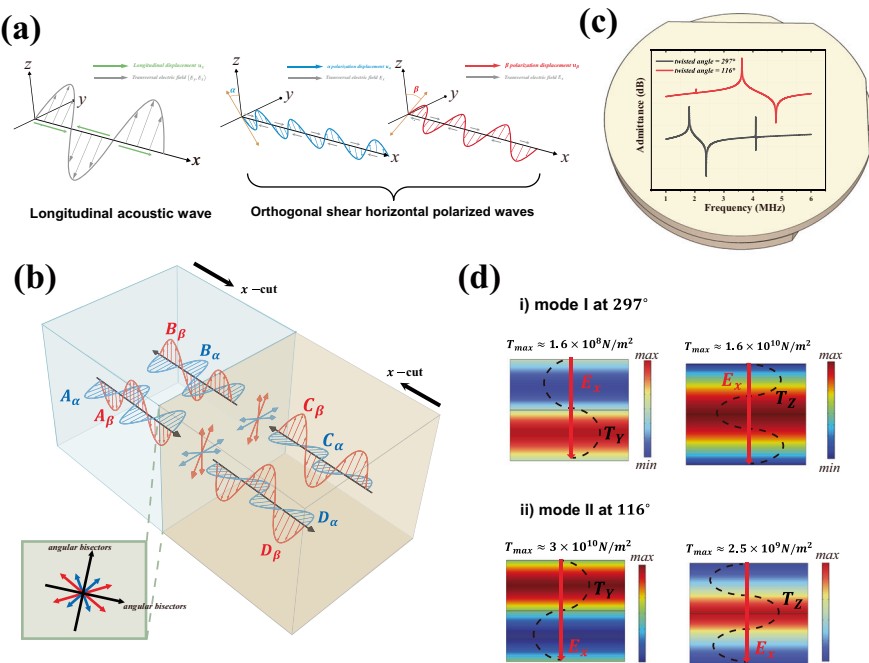

**Fig. 3 | Schematic of superposition of piezoelectrically coupled acoustic waves propagating along x-crystal axis in bulk LN and definition of angular bisectors coordinate system. a** Acoustic waves solved by Eqs. (1) and (2), arrows in green along ±x-axis in (i) represent the longitudinal acoustic displacement ($u_x$), and the coupled transversal electric fields ($E_y$, $E_z$) are shown in gray, noted that the magnetic field components in (i) quasi-longitudinal acoustic wave were not shown. The two piezoelectrical stiffened SH waves ($u_y$, $u_z$) in (ii) and (iii) were in red and blue, respectively, while the coupled longitudinal electric field ($E_x$) was marked as arrows along ±x-axis in gray. **b** Superposition of partial waves in upper and lower LN layers, the coordinate system consists of two angular bisectors between polarization directions of all partial waves were defined in the front view of interface at lower left quarter. **c** FEM-simulated admittance curves at 116 degrees and 297 degrees. **d** Distributions of components of stress fields which projected along the two angular bisectors, the black dashed line indicates symmetric and anti-symmetric.

The quasi-longitudinal acoustic mode and the two piezoelectric stiffened SH modes can be excited respectively if the relevant polarized electric fields were applied, more specifically, utilizing $yz$-inplane polarized electric fields for quasi-longitudinal wave while utilizing $E_x$ for SH waves. In this work we choose to excite the two SH modes propagating along $X$ axis of crystal, so the longitudinal electric field $E_x$ can be obtained by adding alternating electrical signals at upper electrode terminal while taking the bottom electrode as grounding terminal (Fig. 1a). Once the acoustic SH modes excited, the bilayer LN structure can be treated as an acoustical resonant cavity since the SH-type polarized acoustic waves cannot propagate in the air. Furthermore, if the frequency of the electrical excitation signal corresponds to the intrinsic standing modes, a pair of resonance and anti-resonance peaks can be observed in electrical terminal responses test (admittance).

The full knowledge of fields distributions and polarization characteristics, especially of stress and electric fields in ITBLN is the fundamental basic for energy coupling efficiency calculation and adjustment of modes resonant responses. Using partial wave method[29], in which the fields in ITBLN cavity can be constructed as superposition of propagating SH waves solved by Eqs. (1) and (2). As shown in Fig. 3b, the total fields distributions are determined by the amplitudes of those partial waves marked as $A_{\alpha,\beta}$ and $B_{\alpha,\beta}$ in the upper LN, as well as $C_{\alpha,\beta}$ and $D_{\alpha,\beta}$ in the lower LN. The subscripts $\alpha$ and $\beta$ indicate the two orthogonally polarized SH waves (see Supplementary Note 2). The polarization of each wave in upper and lower LN layers are decided through the twisted angle. Reflection and refraction coefficients of those waves at boundaries between different mediums (air and LN layers) must be satisfied and thus the determination of relations of those amplitudes can be obtained and the dispersion of eigenfrequencies and the fields distribution can be derived (see Supplementary Note 2).

To be noted that the analysis can be simplified if the thickness of two LN layers is the same, since the symmetry and anti-symmetry of superposition of partial waves in the resonant cavity allow $A_{\alpha,\beta} = \pm C_{\alpha,\beta}$ as well as $B_{\alpha,\beta} = \pm D_{\alpha,\beta}$, which reduce the determinant from the fourth order into the second order (see Supplementary Note 2). In addition, resonant modes in ITBLN cavity with same thickness of two piezoelectric layers would exhibit symmetric and antisymmetric fields distributions if the coordinate system are defined as the angular bisectors (see Fig. 3b) of polarization direction of SH waves in upper and lower LN layers. There are two series of acoustic resonant modes distinguished by symmetric types: one has symmetric distribution of stress projected along one of the angular bisectors while having antisymmetric distribution of stress projected along the other bisector, another series exhibit the exactly opposite symmetric characteristics. Furthermore, the total polarization of symmetric resonant modes does not keep static along X-crystal axis since the two SH waves for superposition have different phase velocities, or to say having different wavenumbers. To verify the whole fields, we have simulated the electrical responses (admittance curves) of the ITBLN and the stress model shapes corresponding to the resonant frequencies using FEM methods, as shown in Fig. 3c, d. It is evident that only one series of modes can be excited, whose projected symmetrically distributed stress is along the inversed crystal axis, that is the Z-crystal axis in this work. The reason would be discussed later in Section III.

### Determination of optimal angle for electromechanical coupling coefficient

Based on the views of fields distribution, the electromechanical coupling coefficient $K^2$ measuring the energy conversion efficiency can be calculated using Berlincourt equations[30]:

$$K^2 = \frac{U_m^2}{U_e U_d} \qquad (4)$$

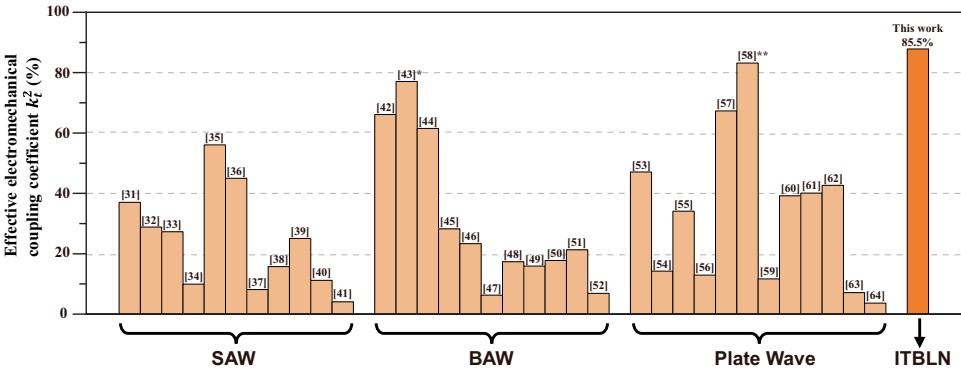

**Fig. 4 | Comparison of effective electromechanical coupling coefficient $k_t^2$ between this work and the state-of-the-art reported acoustic resonators.** *The formula for calculating $k_t^2$ in ref. 43 was $k_t^2 = \pi f_r/2f_a / \tan(\frac{\pi f_r}{2f_a})$, which was different with formula used in this work and the others [$k_t^2 = \pi^2/8\,(f_a^2 - f_r^2)/|f_r^2|$]. According to

the result in ref. 43, which is 43%, the ratio of $f_r/f_a$ was 0.7869, and the unified result in this figure should be 75.9%. **The resonator in ref. 58 was fabricated on the bulk Y27.5 Lithium Niobate substrate, on which the resonate frequency was on the megahertz scale (MHz).

where the mutual energy $U_m = 1/4 \int (\mathbf{T} : \mathbf{d} \cdot \mathbf{E} + \mathbf{E} \cdot \mathbf{d} : \mathbf{T})\,dV$ with $\mathbf{d}$ represents the piezoelectric strain constants, the elastic energy $U_e = 1/2 \int (\mathbf{T} : \mathbf{s}^E : \mathbf{T})\,dV$ with $\mathbf{s}^E$ represents the compliance constants and the electric energy $U_d = 1/2 \int (\mathbf{E} \cdot \boldsymbol{\varepsilon}^S \cdot \mathbf{E})\,dV$, noted that the integral covers the volume of the entire cavity $V$.

As for ITBLN in this work, component form of Eq. (3) was derived as $U_m = 1/4 \int 2E_x(T_5 d_{15} + T_6 d_{16})\,dV$ while $U_e = 1/2 \int (s_{55}T_5^2 + s_{66}T_6^2 + 2s_{56}T_5 T_6)\,dV$ and $U_d = 1/2 \int (\varepsilon_{11}E_x^2)\,dV$. It is worth mentioning that the integral needs to be calculated respectively in upper and lower LN layers since its coordinate crystal system is different. To derive the optimal angle of excitation of the two lowest order modes, we rewrite the Eq. (3) into the structural coordinate system defined by the angular bisectors as discussed before with the following equations: $T_5^{up} = T_Z \cos\theta + T_Y \sin\theta$, $T_6^{up} = T_Z \sin\theta - T_Y \cos\theta$, $T_5^{down} = T_Z \cos\theta - T_Y \sin\theta$ and $T_6^{down} = T_Z \sin\theta + T_Y \cos\theta$, where the superscript "up" and "down" indicates the stress in upper and lower layers, respectively. For simplicity, only $U_m$ was given here:

$$
\begin{aligned}
U_m = &\frac{1}{2} \int_{V^{up}+V^{down}} E_x T_Z (d_{15}\cos\theta + d_{16}\sin\theta)\,dV \\
&+ \frac{1}{2} \int_{V^{up}} E_x T_Y (d_{15}\sin\theta - d_{16}\cos\theta)\,dV \\
&+ \frac{1}{2} \int_{V^{down}} E_x T_Y (-d_{15}\sin\theta + d_{16}\cos\theta)\,dV
\end{aligned}
\tag{5}
$$

where $\theta$ is clockwise defined half twisted angle, $T_Y$ and $T_Z$ are stress components projected along bisectors and related to stress components in crystal system in both LN layers with variables transformations (See Supplementary Note 2).

Noted that it is different between integral of $T_Y$ and $T_Z$ since we choose Z as the inversing axis before twisting. When the thickness of two LN layers equals to each other, one series of eigenmodes solved in section II characterized by symmetric distribution of $T_Y$ and antisymmetric distributio of $T_Z$ can not be excited as indicated by Eq. (4). This is because $T_Y$ is exactly the same in the second and third terms in right hand of Eq. (4), leading the total integral of $T_Y$ across $V^{up}$ and $V^{down}$ to be zero. At the same time, this mode has an anti-symmetric distribution of $T_Z$, leading a zero result of first term in the right hand of Eq. (4) either. On the other hand, a mode has anti-symmetric distribution of $T_Y$ as well as symmetric distribution of $T_Z$ would results in a non-zero value of $U_m$. Thus, only one series of mode which have proper symmetry type can be excited and observed.

The coupling coefficients of the two lowest modes of excitable series can be tuned by twisted angle $2\theta$. Furthermore, due to the

approximation of constant $E_x$ and the stress polarization mainly along Z-axis for the first modes and Y-axis for the second modes, the optimal angle for minimum coupling for the first modes as well as maximum coupling for the second modes can be evaluated by $d_{15}\cos\theta + d_{16}\sin\theta = 0$, which is about 116 degrees in LN crystal. Similarly, the maximum for the first and the minimum for the second can be evaluated by $d_{15}\sin\theta - d_{16}\cos\theta = 0$, being about 296 degrees in LN crystal, which fits well with simulated results discussed above.

A comparison between this work and the state-of-the-art acoustic resonators is shown in Fig. 4[31–64], in which the level of couplings of piezoelectricity is represented by the effective electromechanical coupling coefficient $k_t^2$. The value of $k_t^2$ in this work has exceeded 80%, reaching 85.5%, indicating the incredible large coupling and efficiency of energy transformation. Noted that the transmission zeroes of a ladder-topological acoustic filter are determined by the resonant frequency and anti-resonate frequency of its series and parallel resonators. Therefore, since the distance of two frequency points is characterized by the electromechanical coupling coefficient, it is necessary to achieve larger piezoelectric coupling for wider bands.

## Discussion

We have reported an inversed and twisted bilayer LN structure, ITBLN, which can piezoelectrically excite two shear-horizontal polarized acoustic standing modes with large electromechanical coupling coefficient. This coupling coefficient can be adjusted and the corresponding polarization can be modulated by changing the rotation angle or the thickness ratio of two layers. Confirmed by experimental results, the max of $k_t^2$ has reached 85.5% at rotation angle of 111 degrees when the two layers have the same thickness. We also compared our result with $k_t^2$ of the state-of-the-art acoustic resonators, indicating the largest $k_t^2$ in this work and expectation of breaking the limitation of materials constants of single piezoelectric layer. The demonstrated results are promising for ultra-wideband acoustic filters, and open up more possibilities for next generation wireless communication technologies with higher data capacity.

The characteristic of polarization and mechanism of modulation of piezoelectrically excited standing acoustic modes in ITBLN was analyzed by combination of theory of acoustic fields and finite element simulations. Basic from the partial wave methods, the standing modes in ITBLN can be derived by symmetrically or anti-symmetrically superposition of piezoelectric stiffened SH waves in LN. The Berlincourt equations were used to calculate the optimal rotation angle for max of $k_t^2$ with help of symmetric analysis and simulated distributions of stress fields, the results shows that the theoretical optimal angle is

116 degrees for one mode and 296 degrees for the other, showing well agreement with experiments.

## Methods

### Sample preparation

The preparation process of the ITBLN samples starts with two pairs of double-polished X-cut LN wafers with the thickness of 500 μm and the diameter of 4 inches. Firstly, the LN wafers were firmly bonded without any intermediate or buffer layers through the plasma activated bonding technique. As shown in the right side of Fig. 1c, the twisted angles were set to 35 degrees and 111 degrees. The red and yellow arrows represent the crystal Z-axis of LN wafers. Then the bilayer ITBLN bonding pairs were annealed to enhance the bonding strength. Subsequently, the both sides of ITBLN samples were deposited 500-nm aluminum as electrodes using electron beam evaporation process. As a comparison, the sample with single layer LN slab was also fabricated.

### Admittance measurement

The fabricated samples were mounted on print circuit boards and electrically connected with a Keysight E5071C vector network analyzer through the bonding wires and the coaxial cable to measure the electrical performance. The measured responses were fitted and the $k_t^2$ of resonances was extracted using the multi-resonance modified Butterworth–Van Dyke (MBVD) model[65].

## Data availability

The data for the figures generated in this study have been deposited in the Figshare database under accession code https://doi.org/10.6084/m9.figshare.25662933.

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

## Acknowledgements
This work was supported in part by the National Natural Science Foun-dation of China (62293521, 62293524, 62204252, 62231023, 12361161667), the Youth Innovation Promotion Association CAS, the Young Elite Scientists Sponsorship Program by CAST (2022QNRC001), and the Shanghai Rising-Star Program (23QB1405300).

## Author contributions
H.Y. performed the theoretical analysis, analyzed the data and wrote the initial manuscript under the supervision of S.Z. and with additional support by P.Z. and C.H.; P.Z. also prepared the samples, performed the experiments and measured data of admittance responses with assis-tance from L.Z.; X.F. and H.Y. preformed FEM simulations. D.L. prepared the schematic figures under the supervision of H.Y. and S.Z.; H.C. and X.O. provided guidance throughout the project. S.Z., H.C., and X.O. supervised all aspects of it.

## Competing interests
The authors declare no competing interests.
