## [Peer Review File · Nature Communications]

Twist piezoelectricity: giant electromechanical coupling in magic-angle twisted bilayer LiNbO₃REVIEWER COMMENTS

Reviewer #1 (Remarks to the Author):

This paper presents twist piezoelectricity as a paradigm to modulate polarization and electromechanical coupling by twisting precisely the stacked lithium niobate (LN) slabs due to the interlayer coupling effect. The topic is interesting, and the paper is well written. However, the following critical issues should be sloven, before I recommend this paper to be published.

- 1.The materials properties of LiNbO₃ should be given, such as the electric constants and the piezoelectric constants.
- 2.Are the material properties of LiNbO₃ orthotopic?
- 3.Does the interface effect of the bilayer is neglected? Why?
- 4.More experimental details on how to deal with the interface should be given.
- 5.In abstract, it shows that "...unveiled is unveiled at a pair of magic angles of 111 and 291°, reaching 85.5%". However, Experimental date corresponding to the magic angle of 291° is not shown in the paper. How the authors determine that when the magic angle is 291°, the effective electromechanical coupling coefficient can reach to 85.5% .
- 6.In figure 4, is the effective electromechanical coupling coefficient of this work predicted by theoretical model? Which is the same as the experimental date shown in figure 1. The error between the theoretical model and the experiment is zero?
- 7.Please check Eq.(1). The velocity vector v should be instead by displacement field vector u .
8. In equation (1) the operation of ":" should be defined. u is missed in the first term.
- 9.Some reference should be given for Eq.(1).
- 10.In Eq.(2) , the velocity vector v should be instead by displacement field vector u .
11. cE is a fourth order tensor, e is the third order tensor. These parameters should be "Bold"
- 12.Line 169 and 170, the expressions of U_m , U_e and U_d are wrong since coefficients are missed.
- 13.Some details on how to derive Eq(S9) and (S10) should be given.
- 14.Check Eq. (S10).
- 15.The operation " ∇ " should be defined
- 16.Details on the FEM simulation should be given in the supplementary field. It is convenient for the work to be reproduced.
- 17.Why neglecting the piezoelectricity after Eq.(S33)? Does the effect of piezoelectricity is small? The reasons for neglecting the piezoelectrical term should be explained clearly.
- 18.As shown in figure S3, in the coordinate angular bisectors systems, the relationship between the corresponding shear horizontal stress(T_5 , T_6) and T_z and T_y is not clearly.

Reviewer #2 (Remarks to the Author):

Two layers having equal thickness of LiNbO₃ are stacked together with twisted angle of X cut. Experimental as well as theoretical results are presented. It is observed that at angle of 111oC, K value is maximum and there is no beta phase. Results are compared with other existing devices also. It is clearly sowed that present results are very much superior to others. There are some points are observed.

1. The solution of equation 1 and 2 given in manuscripts is not clear. Is it done by analytical or FEM or software?
2. Figure 6 is not explained clearly in supplementary file.
3. Diameter of film is missing, only thickness is given.
4. Actual experiment set up photo or schematic is missing.
5. Figure S2 in supplementary, Is fig experimental or theoretical?.
6. For each thickness, there may be different angle of optimization. At present case, for equal thickness, it is 111. Can it be verified or justified?
7. The formulation gives in supplementary more useful to understand the paper. If possible, give some of relevant equations in manuscripts or at least cite about supplementary information in the manuscripts for detailed derivation.

Reviewer #3 (Remarks to the Author):

This work constructs a bilayer LiNbO₃ with inverted polarization and twisted angle, and shows an effective electromechanical coupling coefficient of 85.5%.

The work is essentially a classical wave propagation problem in a layered structure, and the phenomenon is well understood from superposition point of view, as authors articulated in the manuscript. I thus fail to see the conceptual advances that were demonstrated in true twistrionics. The manuscript claims giant piezoelectricity and modulation of polarization. Neither of these claims are demonstrated at the material level, and the effective electromechanical coupling coefficient is a result of superposition of waves in two layers, which cancels spurious peaks in the single layer, I believe. The resulting 85.5% coupling coefficient itself is comparable to the corresponding value in single layer as well as state of art devices, not that remarkable as claimed.

Peer Review

**Twist piezoelectricity: giant electromechanical coupling in
magic-angle twisted bilayer LiNbO₃**

Response to the reviewers

We thank all the reviewers for their careful reading of the manuscript and their insightful comments. We have incorporated our responses into the revised manuscript as appropriate. Below please find our point-by-point response to the detailed comments of the reviewers.

Comments of the Reviewer #1

This paper presents twist piezoelectricity as a paradigm to modulate polarization and electromechanical coupling by twisting precisely the stacked lithium niobate (LN) slabs due to the interlayer coupling effect.

Re: We thank the Reviewer #1 for the concise summary.

The topic is interesting, and the paper is well written. However, the following critical issues should be sloven, before I recommend this paper to be published.

Re: We expressed our gratitude to Reviewer #1 for the approval to the topic of our work and positive evaluation for the writings of the paper. Below please find our point-by-point response to the concerns of the reviewer.

Questions of the Reviewer #1

Q1. The materials properties of LiNbO₃ should be given, such as the electric constants and the piezoelectric constants.

Re: We appreciate the reviewer's very good suggestion. The materials properties of LiNbO₃ we used in our paper are listed as follows:

Elastic constants	Values ($10^9 N/m^2$)
C_{11}^E	203
C_{12}^E	53.0
C_{13}^E	75.0
C_{14}^E	9.0
C_{33}^E	245
C_{44}^E	60.0
Piezoelectric stress constants	Values (C/m^2)
e_{15}	3.7
e_{22}	2.5
e_{31}	0.2
e_{33}	1.3
Permittivity	Values ($10^{-9}F/m$)
ϵ_{11}	0.390
ϵ_{33}	0.257

Parameters are cited from: Warner, A.W.; Onoe, M.; Coquin, G.A. Determination of Elastic and

Piezoelectric Constants for Crystals in Class (3m). J. Acoust. Soc. Am. 1967, 42, 1223–1231.

We have added the materials properties of LiNbO₃ into the revised Supplementary Materials (Note 4. FEM simulation of ITBLN resonant cavity) in blue, please kindly check it.

Q2. Are the material properties of LiNbO₃ orthotropic?

Re: Thanks to the reviewer. The material properties of LiNbO₃ are not orthotropic, since an orthotropic material properties require the elastic constants taking form of:

$$\begin{bmatrix} c_{11} & c_{12} & c_{13} & 0 & 0 & 0 \\ c_{12} & c_{11} & c_{23} & 0 & 0 & 0 \\ c_{13} & c_{23} & c_{33} & 0 & 0 & 0 \\ 0 & 0 & 0 & c_{44} & 0 & 0 \\ 0 & 0 & 0 & 0 & c_{55} & 0 \\ 0 & 0 & 0 & 0 & 0 & c_{66} \end{bmatrix}$$

which validates for orthorhombic crystal and has 9 independent constants, while LiNbO₃ belongs to trigonal symmetry point group $3m$, with 6 independent constants and exhibits different form of elastic (as shown in Eq(R27) in our response to Question 13).

Q3. Does the interface effect of the bilayer is neglected? Why?

Figure R1 (a) XTEM image of the interface of bilayer LiNbO₃ plate. (b) Nb and (c) O element distribution at the interface region.

Re: Thanks to the reviewer for the question. We have characterized the interface of the bonded bilayer LiNbO₃ using cross-sectional transmission electron microscope (XTEM) and energy disperse spectroscopy (EDS), and the results are shown below. In Fig. R1(a), the interface between

the upper and lower LiNbO₃ plates is clearly visible, showing a direct material transition without any intermediate or buffer layers. The atomic arrangement of the lower layer is clear due to the formation of strong electron interference. In contrast, the image of the atomic arrangement of the upper layer is blurred because of the twisted angle in the bilayer structure. In other words, it is difficult to see the atomic arrangement of different crystal orientations at the same time. Fig. R1(b) and (c) show the distribution of Nb and O elements, respectively, confirming that both the upper and lower materials are LiNbO₃. **Therefore, the partial waves in the two layers satisfy the continuous boundary condition at the interface**, and the symmetry properties induced by twisting is retained.

We have added the above figures and corresponding consideration of interface effect into the revised Supplementary Materials (Note 6. Experimental setup for measurement of ITBLN acoustic resonators) in blue, please kindly check them.

Q4. More experimental details on how to deal with the interface should be given.

Re: Thanks to the reviewer for the valuable suggestions. Due to the advanced wafer bonding technique, two piezoelectric layers with different crystal orientations are firmly bonded together without any intermediate or buffer layers. The characterization results in the previous reply can prove it. Therefore, the boundary condition at the interface in the experimental sample is continuous, which is consistent with that in the simulations and numerical calculations.

The description of the interface considerations has been added in blue to the revised manuscript: "Firstly, the LN wafers were firmly bonded without any intermediate or buffer layers through the plasma activated bonding technique."

Please kindly check them in the Methods (line 221).

Q5. In abstract, it shows that "...unveiled is unveiled at a pair of magic angles of 111° and 291°, reaching 85.5%". However, Experimental data corresponding to the magic angle of 291° is not shown in the paper. How the authors determine that when the magic angle is 291°, the effective electromechanical coupling coefficient can reach to 85.5%.

Re: We gratefully thank the reviewer for pointing out this question, and apologize for the loose description. In the original manuscript, the magic angles of 297° and 117° have been verified by the FEM simulation, and we predicted that there would be a phase difference of 180° between the two angles by theoretical analysis. In this case, when the magic angle of 111° was demonstrated by the experiment (Fig. 1), another angle was supposed to be 291°. Following the suggestion of the reviewer, we fabricated the ITBLN sample with twisted angle of about 291° recently. The measured k_t^2 of the mode I is about 80%. The inconsistency between the two experiments may be caused by the different piezoelectric constants of different LiNbO₃ batches.

In order to be more rigorous, we have revised the abstract in blue from

"...is unveiled at a pair of magic angles of 111° and 291°, reaching 85.5%"

to

"... is unveiled at magic angle of 111°, reaching 85.5%".

Please check it in the revised manuscript (line 23).

Q6. In figure 4, is the effective electromechanical coupling coefficient of this work predicted by theoretical model? Which is the same as the experimental date shown in figure 1. The error between the theoretical model and the experiment is zero?

Re: Thanks to the reviewer for pointing out this question. The effective electromechanical coupling coefficient of this work in Fig. 4 is calculated by experiments. The value of 85.5% is exactly the result from measured admittance curves of 111° ITBLN, plotted in Fig. 1 of the manuscript.

Furthermore, we have plotted the calculated values of the effective electromechanical coupling coefficient as well as the resonant frequency from the admittance curves simulated by FEM method, as Fig. 2 in the manuscript. The maximum values of the effective electromechanical coupling coefficient in FEM simulations was about 93.4% at 118° being the optimal angle and 93.5% at 298° being the optimal angle. The error of values of both coupling coefficients and optimal angles are mainly due to the errors of the piezoelectric constants of LiNbO₃, which were both used in FEM simulation for admittance and calculations/searching of optimal angles.

Q7. Please check Eq.(1). The velocity vector \mathbf{v} should be instead by displacement field vector \mathbf{u} .

Re: We gratefully thank the reviewer for pointing out this question, and apologize for the mistake that fields variables $\vec{\mathbf{v}}$ or $\vec{\mathbf{u}}$ was missed in the first term. Following the suggestion of the reviewer, we check Eq(1) by providing two procedures of derivation, in which the one takes the velocity vector $\vec{\mathbf{v}}$ as fields variable, and the other replaces $\vec{\mathbf{v}}$ with the displacement field vector $\vec{\mathbf{u}}$.

In a piezoelectric crystal, the acoustic fields equations:

$$\nabla \cdot \mathbf{T} = \rho \frac{\partial \vec{\mathbf{v}}}{\partial t} \quad (\text{R1})$$

$$\nabla_s \vec{\mathbf{v}} = \frac{\partial \mathbf{S}}{\partial t} \quad (\text{R2})$$

where \mathbf{T} is stress tensor, $\vec{\mathbf{v}}$ is the particle velocity defined as first partial derivative of particle displacement with respect to time $\partial \vec{\mathbf{u}}/\partial t$, \mathbf{S} is strain tensor, and the electromagnetic fields equations:

$$\nabla \times \vec{\mathbf{E}} = -\frac{\partial \vec{\mathbf{B}}}{\partial t} \quad (\text{R3})$$

$$\nabla \times \vec{\mathbf{H}} = \frac{\partial \vec{\mathbf{D}}}{\partial t} \quad (\text{R4})$$

where $\vec{\mathbf{E}}$ is electric fields, $\vec{\mathbf{B}}$ is magnetic induction intensity fields, $\vec{\mathbf{H}}$ is magnetic strength fields and $\vec{\mathbf{D}}$ is electric displacement, are coupled through the piezoelectric constitutive relations, which can be given in strain-charge form:

$$\vec{\mathbf{D}} = \boldsymbol{\epsilon}^T \cdot \vec{\mathbf{E}} + \mathbf{d} : \mathbf{T} \quad (\text{R5})$$

$$\mathbf{S} = \mathbf{d} \cdot \vec{\mathbf{E}} + \mathbf{s}^E : \mathbf{T} \quad (\text{R6})$$

where $\boldsymbol{\epsilon}^T$ is permittivity at zero or constant stress, \mathbf{d} is piezoelectric strain constants, \mathbf{s}^E is compliance constants, as well as in stress-charge form:

$$\vec{D} = \epsilon^S \cdot \vec{E} + \mathbf{e} : \mathbf{S} \quad (\text{R7})$$

$$\mathbf{T} = -\mathbf{e} \cdot \vec{E} + \mathbf{c}^E : \mathbf{S} \quad (\text{R8})$$

where ϵ^S is permittivity at zero or constant strain, \mathbf{e} is piezoelectric stress constants, \mathbf{c}^E is elastic constants at zero or constant electric field. Form Eq(R2), with multiplication of \mathbf{c}^E , we obtain:

$$\mathbf{c}^E : \nabla_s \vec{v} = \mathbf{c}^E : \frac{\partial \mathbf{S}}{\partial t} \quad (\text{R9})$$

Simultaneously, the time derivative of Eq(R8) gives:

$$\frac{\partial \mathbf{T}}{\partial t} = -\mathbf{e} \cdot \frac{\partial \vec{E}}{\partial t} + \mathbf{c}^E : \frac{\partial \mathbf{S}}{\partial t} \quad (\text{R10})$$

Thus, substituting Eq(R9) into (R10), we obtain:

$$\frac{\partial \mathbf{T}}{\partial t} = \mathbf{c}^E : \nabla_s \vec{v} - \mathbf{e} \cdot \frac{\partial \vec{E}}{\partial t} \quad (\text{R11})$$

On the other hand, the time derivative of Eq(R1) gives:

$$\nabla \cdot \frac{\partial \mathbf{T}}{\partial t} = \rho \frac{\partial^2 \vec{v}}{\partial t^2} \quad (\text{R12})$$

Combined with Eq(R11) and (R12), the stress \mathbf{T} can be eliminated form acoustic equations:

$$\nabla \cdot \mathbf{c}^E : \nabla_s \vec{v} - \rho \frac{\partial^2 \vec{v}}{\partial t^2} = \nabla \cdot \mathbf{e} \cdot \frac{\partial \vec{E}}{\partial t} \quad (\text{R13})$$

Next we choose to eliminate the magnetic field form the electromagnetic fields equations, the first step is taking the curl of Eq(R3), expressed as:

$$\nabla \times \nabla \times \vec{E} = -\nabla \times \frac{\partial \vec{B}}{\partial t} = -\mu_0 \frac{\partial}{\partial t} (\nabla \times \vec{H}) \quad (\text{R14})$$

where the second equivalence utilized the relationship of $\vec{B} = \mu_0 \vec{H}$. Then substituting Eq(R4) and (R7) into Eq(R14), we obtain:

$$-\nabla \times \nabla \times \vec{E} = \mu_0 \frac{\partial^2 \vec{D}}{\partial t^2} = \mu_0 \epsilon^S \frac{\partial^2 \vec{E}}{\partial t^2} + \mu_0 \mathbf{e} : \frac{\partial^2 \mathbf{S}}{\partial t^2} \quad (\text{R15})$$

From Eq(R2), the time derivative of strain \mathbf{S} can be expressed by the velocity \vec{v} , and the final expression of field equation is:

$$\nabla \times \nabla \times \vec{E} + \mu_0 \epsilon^S \frac{\partial^2 \vec{E}}{\partial t^2} = -\mu_0 \mathbf{e} : \frac{\partial}{\partial t} \nabla_s \vec{v} \quad (\text{R16})$$

Equation (R13) and (R16) is the final formalism of piezoelectric field equations.

Next, if choosing the displacement field \vec{u} instead of velocity field \vec{v} , the acoustic field equations (R1) and (R2) should be rewritten as:

$$\nabla \cdot \mathbf{T} = \rho \frac{\partial^2 \vec{\mathbf{u}}}{\partial t^2} \quad (\text{R17})$$

$$\nabla_s \vec{\mathbf{u}} = \mathbf{S} \quad (\text{R18})$$

Multiplication of (R18) by \mathbf{c}^E leads to:

$$\mathbf{c}^E : \nabla_s \vec{\mathbf{u}} = \mathbf{c}^E : \mathbf{S} \quad (\text{R19})$$

Next, substitution (R19) into (R8) gives:

$$\mathbf{T} = -\mathbf{e} \cdot \vec{\mathbf{E}} + \mathbf{c}^E : \nabla_s \vec{\mathbf{u}} \quad (\text{R20})$$

Thus, the acoustic field equation corresponding to (R13) with $\vec{\mathbf{u}}$ instead of $\vec{\mathbf{v}}$ can be obtained:

$$\nabla \cdot \mathbf{c}^E : \nabla_s \vec{\mathbf{u}} - \rho \frac{\partial^2 \vec{\mathbf{u}}}{\partial t^2} = \nabla \cdot \mathbf{e} \cdot \vec{\mathbf{E}} \quad (\text{R21})$$

As for the electromagnetic fields equation, the procedure of eliminating the magnetic field is same as derivation from Eq(R3) to (R15). The only difference is substituting Eq(R18) instead of (R2) into Eq(R15), which gives the final formalism:

$$\nabla \times \nabla \times \vec{\mathbf{E}} + \mu_0 \epsilon^S \frac{\partial^2 \vec{\mathbf{E}}}{\partial t^2} = -\mu_0 \mathbf{e} : \frac{\partial^2}{\partial t^2} \nabla_s \vec{\mathbf{u}} \quad (\text{R22})$$

As a conclusion, both piezoelectric field equations derived above are relisted here:

Velocity filed $\vec{\mathbf{v}}$ as filed variables	Displacement filed $\vec{\mathbf{u}}$ as filed variables
$\begin{cases} \nabla \cdot \mathbf{c}^E : \nabla_s \vec{\mathbf{v}} - \rho \frac{\partial^2 \vec{\mathbf{v}}}{\partial t^2} = \nabla \cdot \mathbf{e} \cdot \frac{\partial \vec{\mathbf{E}}}{\partial t} \\ \nabla \times \nabla \times \vec{\mathbf{E}} + \mu_0 \epsilon^S \frac{\partial^2 \vec{\mathbf{E}}}{\partial t^2} = -\mu_0 \mathbf{e} : \frac{\partial}{\partial t} \nabla_s \vec{\mathbf{v}} \end{cases}$	$\begin{cases} \nabla \cdot \mathbf{c}^E : \nabla_s \vec{\mathbf{u}} - \rho \frac{\partial^2 \vec{\mathbf{u}}}{\partial t^2} = \nabla \cdot \mathbf{e} \cdot \vec{\mathbf{E}} \\ \nabla \times \nabla \times \vec{\mathbf{E}} + \mu_0 \epsilon^S \frac{\partial^2 \vec{\mathbf{E}}}{\partial t^2} = -\mu_0 \mathbf{e} : \frac{\partial^2}{\partial t^2} \nabla_s \vec{\mathbf{u}} \end{cases}$

Please kindly check the equation (1) and (2) in the revised manuscript (line 117 and 118) and the revised Supplemental Material (part A of Note 1. Derivation of piezoelectric coupled acoustic waves in X-cut LiNbO₃ crystal) in blue.

Q8. In equation (1) the operation of “:” should be defined. u is missed in the first term.

Re: We appreciate the reviewer’s very good suggestion, and apologize for the mistake that fields variables $\vec{\mathbf{u}}$ was missed in the first term. The operation of “:” in equations is defined as contraction of tensor.

Please kindly check it in the revised manuscript (line 120 and 121) in blue.

Q9. Some reference should be given for Eq.(1).

Re: We appreciate the reviewer’s good suggestion. Eq(1) comes from the reference [29] cited in the manuscript. More precisely, the formalism of Eq(1) and (2) can be found as equation (8.105) and

(8.106) in [29], respectively. Furthermore, the derivation of Eq(R13) and (R16) in response of question 7 is also origins from the derivation of Eq(8.105) and (8.106) in reference [29]. The only difference is that we treat the piezoelectric medium with lossless and having no sources at the beginning, which means that the body force term \vec{F} , the conduction current density term \vec{J}_c and the source current density \vec{J}_s do not appear in our derivation.

Following the suggestion of the reviewer, we have added the citation of Eq(1) and (2) in the revised manuscript in blue. Please kindly check it in the revised manuscript (line 116).

[29] Auld B. Acoustic Fields and Waves in Solids. Wiley Interscience Pub. 1973; p. 278.

Q10. In Eq.(2) , the velocity vector v should be instead by displacement field vector u .

Re: We appreciate the reviewer's good suggestion. As the final formalism of electromagnetic fields equation, Eq(2) is derived simultaneously with Eq(1), and the derivation was provided before in our response to the question 7. Following the suggestion of the reviewer, we have changed the formalism of Eq(2) into the displacement field vector \vec{u} version, which is Eq(R22):

$$\nabla \times \nabla \times \vec{E} + \mu_0 \epsilon^S \frac{\partial^2 \vec{E}}{\partial t^2} = -\mu_0 \mathbf{e} : \frac{\partial^2}{\partial t^2} \nabla_s \vec{u}$$

Please kindly check the Eq(2) in the revised manuscript (line 118).

Q11. cE is a fourth order tensor, e is the third order tensor. These parameters should be "Bold".

Re: We thank the reviewer for pointing out this issue. The fourth order tensor c^E and third order tensor e in expression of equations and manuscript have been checked with "Bold".

Please kindly check the equation (1) and (2) in the revised manuscript (line 117 and 118) and Note 1 of the revised Supplemental Material.

Q12. Line 169 and 170, the expressions of U_m , U_e and U_d are wrong since coefficients are missed.

Re: Reviewer is right, we missed the coefficients of U_m , U_e and U_d in original manuscript in line 169 and 170. The correct expression should be:

$$U_m = \frac{1}{4} \int 2E_x (T_5 d_{15} + T_6 d_{16}) dV \quad (R23)$$

$$U_e = \frac{1}{2} \int (s_{55} T_5^2 + s_{66} T_6^2 + 2s_{56} T_5 T_6) dV \quad (R24)$$

$$U_d = \frac{1}{2} \int (\epsilon_{11} E_x^2) dV \quad (R25)$$

We have corrected this error in blue. Please kindly check them in the revised manuscript (line 178 and 179).

Q13. Some details on how to derive Eq(S9) and (S10) should be given.

Re: We appreciate the reviewer's good suggestion. To derive Eq(S9) and (S10), some simplifications

need to be performed first. Since the solutions are plane waves propagating along X axis of LiNbO₃, all corresponding components of field should take form of $\exp(j\omega t - jkx)$. Therefore, the operators $\partial/\partial y$ and $\partial/\partial z$ leads to zero during the derivation, as well as the time derivative operator $\partial/\partial t$ leads to $j\omega$.

Before using these simplifications, Eq(R21) should be firstly rewritten in matrix form:

$$\begin{pmatrix} \frac{\partial}{\partial x} & 0 & 0 & 0 & \frac{\partial}{\partial z} & \frac{\partial}{\partial y} \\ 0 & \frac{\partial}{\partial y} & 0 & \frac{\partial}{\partial z} & 0 & \frac{\partial}{\partial x} \\ 0 & 0 & \frac{\partial}{\partial z} & \frac{\partial}{\partial y} & \frac{\partial}{\partial x} & 0 \end{pmatrix} \cdot [c^E] \cdot \begin{pmatrix} \frac{\partial}{\partial x} & 0 & 0 \\ 0 & \frac{\partial}{\partial y} & 0 \\ 0 & 0 & \frac{\partial}{\partial z} \\ 0 & \frac{\partial}{\partial z} & \frac{\partial}{\partial y} \\ \frac{\partial}{\partial z} & 0 & \frac{\partial}{\partial x} \\ \frac{\partial}{\partial y} & \frac{\partial}{\partial x} & 0 \end{pmatrix} \cdot \begin{bmatrix} u_x \\ u_y \\ u_z \end{bmatrix} - \rho \frac{\partial^2}{\partial t^2} \mathbf{I}_{3 \times 3} \cdot \begin{bmatrix} u_x \\ u_y \\ u_z \end{bmatrix} = \begin{pmatrix} \frac{\partial}{\partial x} & 0 & 0 & 0 & \frac{\partial}{\partial z} & \frac{\partial}{\partial y} \\ 0 & \frac{\partial}{\partial y} & 0 & \frac{\partial}{\partial z} & 0 & \frac{\partial}{\partial x} \\ 0 & 0 & \frac{\partial}{\partial z} & \frac{\partial}{\partial y} & \frac{\partial}{\partial x} & 0 \end{pmatrix} \cdot [e] \cdot \begin{bmatrix} E_x \\ E_y \\ E_z \end{bmatrix} \quad (\text{R26})$$

where $[c^E]$ is a 6×6 sized matrix, representing the elastic constants of LN and $[e]$ is a 6×3 sized matrix, representing the piezoelectric constants of LN:

$$[c^E] = \begin{bmatrix} c_{11} & c_{12} & c_{13} & c_{14} & 0 & 0 \\ c_{12} & c_{11} & c_{13} & -c_{14} & 0 & 0 \\ c_{13} & c_{13} & c_{33} & 0 & 0 & 0 \\ c_{14} & -c_{14} & 0 & c_{44} & 0 & 0 \\ 0 & 0 & 0 & 0 & c_{55} & c_{56} \\ 0 & 0 & 0 & 0 & c_{56} & c_{66} \end{bmatrix} \quad (\text{R27})$$

Noted that c_{55} equals to c_{44} , c_{56} equals to c_{14} and c_{66} equals to $(c_{11} - c_{12})/2$.

$$[e] = \begin{bmatrix} 0 & -e_{22} & e_{31} \\ 0 & e_{22} & e_{31} \\ 0 & 0 & e_{33} \\ 0 & e_{15} & 0 \\ e_{15} & 0 & 0 \\ e_{16} & 0 & 0 \end{bmatrix} \quad (\text{R28})$$

Noted that e_{16} equals to $-e_{22}$. Substituting Eq(R27) and (R28) into Eq(R26) under the simplifications discussed above, we obtain:

$$\left\{ \frac{\partial^2}{\partial x^2} \begin{bmatrix} c_{11} & 0 & 0 \\ 0 & c_{66} & c_{56} \\ 0 & c_{56} & c_{55} \end{bmatrix} - \rho \frac{\partial^2}{\partial t^2} \mathbf{I}_{3 \times 3} \right\} \begin{bmatrix} u_x \\ u_y \\ u_z \end{bmatrix} = \frac{\partial}{\partial x} \begin{bmatrix} -e_{22}E_y + e_{31}E_z \\ e_{16}E_x \\ e_{15}E_x \end{bmatrix} \quad (\text{R29})$$

which is the Eq(S14) in the revised Supplementary Materials.

Ons the other hand, the matrix form of Eq(R22) was given as:

$$\begin{pmatrix} 0 & -\frac{\partial}{\partial z} & \frac{\partial}{\partial y} \\ \frac{\partial}{\partial z} & 0 & -\frac{\partial}{\partial x} \\ -\frac{\partial}{\partial y} & \frac{\partial}{\partial x} & 0 \end{pmatrix} \cdot \begin{pmatrix} 0 & -\frac{\partial}{\partial z} & \frac{\partial}{\partial y} \\ \frac{\partial}{\partial z} & 0 & -\frac{\partial}{\partial x} \\ -\frac{\partial}{\partial y} & \frac{\partial}{\partial x} & 0 \end{pmatrix} \cdot \begin{bmatrix} E_x \\ E_y \\ E_z \end{bmatrix} + \mu_0 [\epsilon^S] \cdot \frac{\partial^2}{\partial t^2} \mathbf{I}_{3 \times 3} \cdot \begin{bmatrix} E_x \\ E_y \\ E_z \end{bmatrix} = -\mu_0 \frac{\partial^2}{\partial t^2} [e]^T \begin{pmatrix} \frac{\partial}{\partial x} & 0 & 0 \\ 0 & \frac{\partial}{\partial y} & 0 \\ 0 & 0 & \frac{\partial}{\partial z} \\ 0 & \frac{\partial}{\partial z} & \frac{\partial}{\partial y} \\ \frac{\partial}{\partial z} & 0 & \frac{\partial}{\partial x} \\ \frac{\partial}{\partial y} & \frac{\partial}{\partial x} & 0 \end{pmatrix} \begin{bmatrix} u_x \\ u_y \\ u_z \end{bmatrix} \quad (\text{R30})$$

where $[\epsilon^S]$ is a 3×3 sized matrix, representing the permittivity constants of LN:

$$[\epsilon^S] = \begin{bmatrix} \epsilon_{11} & 0 & 0 \\ 0 & \epsilon_{11} & 0 \\ 0 & 0 & \epsilon_{33} \end{bmatrix} \quad (\text{R31})$$

Substituting Eq(R31) and (R28) into Eq(R30) under the simplifications discussed above, we obtain:

$$\left\{ \begin{pmatrix} 0 & 0 & 0 \\ 0 & \frac{\partial^2}{\partial x^2} & 0 \\ 0 & 0 & \frac{\partial^2}{\partial x^2} \end{pmatrix} - \mu_0 \frac{\partial^2}{\partial t^2} \begin{bmatrix} \epsilon_{11} & 0 & 0 \\ 0 & \epsilon_{11} & 0 \\ 0 & 0 & \epsilon_{33} \end{bmatrix} \right\} \begin{bmatrix} E_x \\ E_y \\ E_z \end{bmatrix} = \mu_0 \frac{\partial^3}{\partial x \partial t^2} \begin{bmatrix} e_{16} u_y + e_{15} u_z \\ -e_{22} u_x \\ e_{31} u_x \end{bmatrix} \quad (\text{R32})$$

which is the Eq(S15) in the revised Supplementary Materials.

Noted that all equations above are derived with displacement vector \vec{u} as filed variables, which is different from the original formalism of Eq(S9) and Eq(S10) in the original Supplementary Materials. To make it more clear, here we also provide derivation details with velocity vector \vec{v} as filed variables.

The matrix form of Eq(R13) and Eq(R16) can be given respectively as follows:

$$\begin{pmatrix} \frac{\partial}{\partial x} & 0 & 0 & 0 & \frac{\partial}{\partial z} & \frac{\partial}{\partial y} \\ 0 & \frac{\partial}{\partial y} & 0 & \frac{\partial}{\partial z} & 0 & \frac{\partial}{\partial x} \\ 0 & 0 & \frac{\partial}{\partial z} & \frac{\partial}{\partial y} & \frac{\partial}{\partial x} & 0 \end{pmatrix} \cdot [c^E] \cdot \begin{pmatrix} \frac{\partial}{\partial x} & 0 & 0 \\ 0 & \frac{\partial}{\partial y} & 0 \\ 0 & 0 & \frac{\partial}{\partial z} \\ 0 & \frac{\partial}{\partial z} & \frac{\partial}{\partial y} \\ \frac{\partial}{\partial z} & 0 & \frac{\partial}{\partial x} \\ \frac{\partial}{\partial y} & \frac{\partial}{\partial x} & 0 \end{pmatrix} \cdot \begin{bmatrix} v_x \\ v_y \\ v_z \end{bmatrix} - \rho \frac{\partial^2}{\partial t^2} \mathbf{I}_{3 \times 3} \cdot \begin{bmatrix} u_x \\ u_y \\ u_z \end{bmatrix} = \begin{pmatrix} \frac{\partial}{\partial x} & 0 & 0 & 0 & \frac{\partial}{\partial z} & \frac{\partial}{\partial y} \\ 0 & \frac{\partial}{\partial y} & 0 & \frac{\partial}{\partial z} & 0 & \frac{\partial}{\partial x} \\ 0 & 0 & \frac{\partial}{\partial z} & \frac{\partial}{\partial y} & \frac{\partial}{\partial x} & 0 \end{pmatrix} \cdot [e] \cdot \frac{\partial}{\partial t} \begin{bmatrix} E_x \\ E_y \\ E_z \end{bmatrix} \quad (\text{R33})$$

$$\begin{pmatrix} 0 & -\frac{\partial}{\partial z} & \frac{\partial}{\partial y} \\ \frac{\partial}{\partial z} & 0 & -\frac{\partial}{\partial x} \\ -\frac{\partial}{\partial y} & \frac{\partial}{\partial x} & 0 \end{pmatrix} \cdot \begin{pmatrix} 0 & -\frac{\partial}{\partial z} & \frac{\partial}{\partial y} \\ \frac{\partial}{\partial z} & 0 & -\frac{\partial}{\partial x} \\ -\frac{\partial}{\partial y} & \frac{\partial}{\partial x} & 0 \end{pmatrix} \cdot \begin{bmatrix} E_x \\ E_y \\ E_z \end{bmatrix} + \mu_0 [\epsilon^S] \cdot \frac{\partial^2}{\partial t^2} \mathbf{I}_{3 \times 3} \cdot \begin{bmatrix} E_x \\ E_y \\ E_z \end{bmatrix} = -\mu_0 \frac{\partial}{\partial t} [e]^\top \begin{pmatrix} \frac{\partial}{\partial x} & 0 & 0 \\ 0 & \frac{\partial}{\partial y} & 0 \\ 0 & 0 & \frac{\partial}{\partial z} \\ 0 & \frac{\partial}{\partial z} & \frac{\partial}{\partial y} \\ \frac{\partial}{\partial z} & 0 & \frac{\partial}{\partial x} \\ \frac{\partial}{\partial y} & \frac{\partial}{\partial x} & 0 \end{pmatrix} \begin{bmatrix} v_x \\ v_y \\ v_z \end{bmatrix} \quad (R34)$$

Substituting Eq(R27), (R28) and (R31) under the simplifications discussed above, we obtain:

$$\left\{ \frac{\partial^2}{\partial x^2} \begin{bmatrix} c_{11} & 0 & 0 \\ 0 & c_{66} & c_{56} \\ 0 & c_{56} & c_{55} \end{bmatrix} - \rho \frac{\partial^2}{\partial t^2} \mathbf{I}_{3 \times 3} \right\} \begin{bmatrix} v_x \\ v_y \\ v_z \end{bmatrix} = \frac{\partial^2}{\partial x \partial t} \begin{bmatrix} -e_{22} E_y + e_{31} E_z \\ e_{16} E_x \\ e_{15} E_x \end{bmatrix} \quad (R35)$$

$$\left\{ \begin{pmatrix} 0 & 0 & 0 \\ 0 & \frac{\partial^2}{\partial x^2} & 0 \\ 0 & 0 & \frac{\partial^2}{\partial x^2} \end{pmatrix} - \mu_0 \frac{\partial^2}{\partial t^2} \begin{bmatrix} \epsilon_{11} & 0 & 0 \\ 0 & \epsilon_{11} & 0 \\ 0 & 0 & \epsilon_{33} \end{bmatrix} \right\} \begin{bmatrix} E_x \\ E_y \\ E_z \end{bmatrix} = \mu_0 \frac{\partial^2}{\partial x \partial t} \begin{bmatrix} e_{16} v_y + e_{15} v_z \\ -e_{22} v_x \\ e_{31} v_x \end{bmatrix} \quad (R36)$$

The details of derivations have been added into the revised Supplementary Materials (Note 1. Derivation of piezoelectric coupled acoustic waves in X-cut LiNbO₃ crystal) in blue. Please kindly check them in the revised Supplementary Materials.

Q14. Check Eq. (S10).

Re: We appreciate the reviewer's good suggestion. We have checked the formalism and derivation of Eq(S10) in the original Supplementary Materials, i.e., Eq(S15) in the revised Supplementary Materials, in our response to Question 13. To be consistent with Eq(1) and (2) in the revised manuscript, we have updated the formalism with displacement type, which is Eq(R35) and (R36).

Please kindly check them in the revised Supplementary Materials (part A of Note 1. Derivation of piezoelectric coupled acoustic waves in X-cut LiNbO₃ crystal).

Q15. The operation “ ∇ ” should be defined.

Re: We appreciate the reviewer's good suggestion. As shown in our response to Question 13, the operation “ ∇ ” used in the derivation and equations can be written as matrix form. According to the variable type of the object of the operator “ ∇ ”, there are three types of calculations need to be defined:

$$\nabla \cdot \rightarrow \begin{pmatrix} \frac{\partial}{\partial x} & 0 & 0 & 0 & \frac{\partial}{\partial z} & \frac{\partial}{\partial y} \\ 0 & \frac{\partial}{\partial y} & 0 & \frac{\partial}{\partial z} & 0 & \frac{\partial}{\partial x} \\ 0 & 0 & \frac{\partial}{\partial z} & \frac{\partial}{\partial y} & \frac{\partial}{\partial x} & 0 \end{pmatrix} \quad (R37)$$

where the operator acted on a symmetric second order tensor (like **S** or **T**) and the result should be

a vector (like \vec{u} or \vec{E}),

$$\nabla \times \rightarrow \begin{pmatrix} 0 & -\frac{\partial}{\partial z} & \frac{\partial}{\partial y} \\ \frac{\partial}{\partial z} & 0 & -\frac{\partial}{\partial x} \\ -\frac{\partial}{\partial y} & \frac{\partial}{\partial x} & 0 \end{pmatrix} \quad (\text{R38})$$

where the operator acted on a vector and the result should also be a vector,

$$\nabla_s \rightarrow \begin{pmatrix} \frac{\partial}{\partial x} & 0 & 0 \\ 0 & \frac{\partial}{\partial y} & 0 \\ 0 & 0 & \frac{\partial}{\partial z} \\ 0 & \frac{\partial}{\partial z} & \frac{\partial}{\partial y} \\ \frac{\partial}{\partial z} & 0 & \frac{\partial}{\partial x} \\ \frac{\partial}{\partial y} & \frac{\partial}{\partial x} & 0 \end{pmatrix} \quad (\text{R39})$$

where the operator acted on a vector and the result should be a symmetric second-order tensor.

We have added the definition above into the revised Supplementary Materials (part B of Note 1). Please kindly check them in the revised Supplementary Materials in blue.

Q16. Details on the FEM simulation should be given in the supplementary field. It is convenient for the work to be reproduced.

Re: We thank the reviewer for pointing out this question. Our FEM simulation was preformed based on the COMSOL Multiphysics. Two layers of LiNbO₃ with size of 2um × 2um wide and 500nm thick were stacked. At the top surface of the upper LiNbO₃ layer and bottom surface of the lower LiNbO₃ layer, two electric terminals with equivalent electrical potential (1V and 0V respectively) were added to simulate the top and bottom electrodes and excite acoustic resonance. Two pairs of periodic boundaries, more precisely as continuous conditions, were added on both sides of layers. Since the structure of twisted bilayer LiNbO₃ requires the head-to-head set of X-axis between the upper and lower layers and exhibits a twisting angle (θ_T), we utilized the Euler angles (α, β, γ) systems in both layers, making ($-\theta_T, -90^\circ, 90^\circ$) for the upper layer and ($0^\circ, -90^\circ, -90^\circ$) for the lower layer. We employ two simulation modules: solid mechanics and electrostatics while the strain and electrostatic fields are coupled together via the piezoelectric effects.

We have added a new note in blue to describe the FEM simulation setups in our work in the revised Supplementary Materials (Note 4. FEM simulation of ITBLN resonant cavity). Please kindly check it the revised Supplementary Materials.

Q17. Why neglecting the piezoelectricity after Eq.(S33)? Does the effect of piezoelectricity is small? The reasons for neglecting the piezoelectrical term should be explained clearly.

Re: We gratefully thank the reviewer for the instructive suggestion. Primarily, we address the complete derivation with consideration of the piezoelectricity, and compare the formalism of key equations with/without neglecting the piezoelectricity in the manuscript and Supplementary Materials.

The complete derivation begins with the redefinition of acoustic impedance of α and β polarized modes (Z_α and Z_β , respectively). Firstly, with consideration of piezoelectric term $-e_{ij}E_j$, the stress of α and β modes can be calculated as:

$$T_I = c_{IJ}\nabla_{Ji}u_i - e_{Ij}E_j \quad (\text{R40})$$

Since either α and β exhibits shear-horizontal polarization (u_y and u_z) and merely couples with longitudinal electric fields (E_x), substituting Eq(R27), (R28) and (R39) into (R40) would lead to nonzero value of T_5 and T_6 as well as the other components remain zero, given as:

$$T_5 = \frac{\partial}{\partial x}(c_{56}u_y + c_{55}u_z) - e_{15}E_x \quad (\text{R41})$$

$$T_6 = \frac{\partial}{\partial x}(c_{66}u_y + c_{56}u_z) - e_{16}E_x \quad (\text{R42})$$

Furthermore, since the curl of the electric fields maintains zero even in time harmonic vibration and thus there is no time harmonic vibrating magnetic fields, the electric displacement fields should be zero indicated by equation (R4). Therefore, the coupled longitudinal electric fields can be calculated as:

$$E_x = -\frac{1}{\epsilon_{11}}(e_{15}S_5 + e_{16}S_6) \quad (\text{R43})$$

On the other hand, relations Eq(R18) between strain \mathbf{S} and displacement $\vec{\mathbf{u}}$ gives:

$$S_5 = \frac{\partial u_z}{\partial x} \quad (\text{R44})$$

$$S_6 = \frac{\partial u_y}{\partial x} \quad (\text{R45})$$

Thus, simultaneously combined with from Eq(R41) to (R44), the stress of α and β modes with consideration of piezoelectricity were obtained:

$$T_5 = \frac{\partial}{\partial x} \left[\left(c_{56} + \frac{e_{15}e_{16}}{\epsilon_{11}} \right) u_y + \left(c_{55} + \frac{e_{15}^2}{\epsilon_{11}} \right) u_z \right] \quad (\text{R46})$$

$$T_6 = \frac{\partial}{\partial x} \left[\left(c_{66} + \frac{e_{16}^2}{\epsilon_{11}} \right) u_y + \left(c_{56} + \frac{e_{15}e_{16}}{\epsilon_{11}} \right) u_z \right] \quad (\text{R47})$$

Secondly, we come back to the dispersion equations derived from Eq(R23) and (R29). For the two shear-horizontal polarized modes, we have:

$$k^2 \begin{bmatrix} c_{66} + \frac{e_{16}^2}{\epsilon_{11}} & c_{56} + \frac{e_{15}e_{16}}{\epsilon_{11}} \\ c_{56} + \frac{e_{15}e_{16}}{\epsilon_{11}} & c_{55} + \frac{e_{15}^2}{\epsilon_{11}} \end{bmatrix} \cdot \begin{bmatrix} u_y \\ u_z \end{bmatrix} = \rho\omega^2 \begin{bmatrix} u_y \\ u_z \end{bmatrix} \quad (\text{R48})$$

The polarization of SH modes needs to be solved. By dividing with u_z at both sides of the Eq(R48), the dispersion equations can be rewritten as:

$$k^2(c'_{66}\gamma + c'_{56}) = \rho\omega^2\gamma \quad (\text{R49})$$

$$k^2(c'_{56}\gamma + c'_{55}) = \rho\omega^2 \quad (\text{R50})$$

where γ denotes the polarization by $\gamma \equiv u_y/u_z$, and coefficients c'_{66} , c'_{56} , and c'_{55} is defined for simplification as $c_{66} + e_{16}^2/\epsilon_{11}$, $c_{56} + e_{15}e_{16}/\epsilon_{11}$ and $c_{55} + e_{15}^2/\epsilon_{11}$, respectively. Combination of Eq(R49) and (R50) gives a quadratic equation of γ :

$$c'_{56}\gamma^2 + (c'_{55} - c'_{66})\gamma - c'_{56} = 0 \quad (\text{R51})$$

Eq(R51) can be also written as:

$$\gamma(c'_{56}\gamma + c'_{55}) = c'_{66}\gamma + c'_{56} \quad (\text{R52})$$

Here we marked the two solutions to Eq(R51) or (R52) as α and β , which corresponds to the α and β modes solved by Eq(R48). Obviously, both α and β must satisfy the equations above.

Thirdly, with help of the polarization equations derived above, the stress and displacement of α and β modes can be related. Considering a wave propagating along +x direction with α or β polarization type, its displacement field can be written as:

$$\vec{\mathbf{u}} = (u_y\hat{\mathbf{y}} + u_z\hat{\mathbf{z}}) \quad (\text{R53})$$

where the harmonic term $\exp(j\omega t - jkx)$ is omitted. Substituting γ into Eq(R53) gives:

$$\vec{\mathbf{u}} = u_z(\gamma\hat{\mathbf{y}} + \hat{\mathbf{z}}) \quad (\text{R54})$$

Meanwhile, according to the projection of stress along x axis:

$$\mathbf{T} \cdot \hat{\mathbf{x}} = T_1\hat{\mathbf{x}} + T_6\hat{\mathbf{y}} + T_5\hat{\mathbf{z}} \quad (\text{R55})$$

Combination with Eq(R46) and (R47), the projection of stress field can be written as:

$$\mathbf{T} \cdot \hat{\mathbf{x}} = -jk[(c'_{66}u_y + c'_{56}u_z)\hat{\mathbf{y}} + (c'_{56}u_y + c'_{55}u_z)\hat{\mathbf{z}}] \quad (\text{R56})$$

Substituting γ into Eq(R56) gives:

$$\mathbf{T} \cdot \hat{\mathbf{x}} = -jku_z(c'_{56}\gamma + c'_{55})(\gamma\hat{\mathbf{y}} + \hat{\mathbf{z}}) \quad (\text{R57})$$

where relation Eq(R52) was used during the derivation. To be noted that the complex form of displacement and stress field involves an imaginary part in expressions, which defines the phase. The actual field should be the real part during the harmonic period. On the other hand, the definition of acoustic impedance Z_γ gives:

$$Z_\gamma = \frac{\rho\omega}{k_\gamma} \quad (\text{R58})$$

where γ equals either to the two solutions (α or β). Therefore, according to Eq(R50), (R54) and (R57), the projection stress of α or β mode can be related with its displacement field:

$$\mathbf{T} \cdot \hat{x} = -j\omega Z_\gamma \vec{\mathbf{u}} = -Z_\gamma \vec{\mathbf{v}} \quad (\text{R59})$$

Furthermore, if the wave propagates along -x direction, its harmonic term should be $\exp(j\omega t + kx)$. Under this condition, the relation should be:

$$\mathbf{T} \cdot \hat{x} = Z_\gamma \vec{\mathbf{v}} \quad (\text{R60})$$

where γ equals either to the two solutions (α or β). Moreover, the phase velocity (calculated as ω/k) of α and β modes can be derived from Eq(R49) or (R50), which provides an expression of the acoustic impedance in terms of material constants:

$$Z_\gamma = \sqrt{\rho(c'_{56}\gamma + c'_{55})} = \sqrt{\rho\gamma^{-1}(c'_{66}\gamma + c'_{56})} \quad (\text{R61})$$

Additionally, neglecting the piezoelectricity during the derivation above gives different but similar structure forms of stress expressions Eq(R46) and (R47), dispersion equations Eq(R48) and acoustic impedance Eq(R61), given respectively as:

$$T_5 = \frac{\partial}{\partial x} (c_{56}u_y + c_{55}u_z) \quad (\text{R62})$$

$$T_6 = \frac{\partial}{\partial x} (c_{66}u_y + c_{56}u_z) \quad (\text{R63})$$

$$k^2 \begin{bmatrix} c_{66} & c_{56} \\ c_{56} & c_{55} \end{bmatrix} \cdot \begin{bmatrix} u_y \\ u_z \end{bmatrix} = \rho\omega^2 \begin{bmatrix} u_y \\ u_z \end{bmatrix} \quad (\text{R64})$$

$$Z_\gamma = \sqrt{\rho(c_{56}\gamma + c_{55})} = \sqrt{\rho\gamma^{-1}(c_{66}\gamma + c_{56})} \quad (\text{R65})$$

Obviously, for a piezoelectric material in this case, **the piezoelectricity can be involved in the derivation by simply replacing the original elastic constants c_{55} , c_{56} and c_{66} with the piezoelectric stiffened elastic constants c'_{55} , c'_{56} and c'_{66} , while all the math and expressions remain the same formalism.** Particularly, the stress expressions of partial waves (from Eq(S62) to (S66), Section II.3 in the revised Supplementary Materials) have been taking the piezoelectricity into consideration by treating the acoustic impedance as Eq(R61) rather than (R65). Thus, the stress distributions in the twisted bilayer LiNbO₃ as a result of superposition of α and β polarized eigenmodes would still exhibit two series of symmetry, guaranteeing that the derivation and calculation of optimal angles for largest effective electromechanical coupling coefficient unperturbed by the piezoelectricity.

On the other hand, piezoelectricity mainly influences the calculation of refraction T and reflection Γ coefficients (Section II.1 in the revised Supplementary Materials), which leads to the change of resonant frequency determined by Eq(S28), Section II.1 in the revised Supplementary

Materials). In fact, we choose FEM simulations to obtain the resonant frequency, but provide the theoretical determinations for a complete and clear physical image of superposition. In our original Supplementary Materials, the piezoelectricity was neglected since the derivations of distributions of stress fields and the discussions on its symmetry characteristics would not be affected by the piezoelectric terms. In order to simplify the equations and give a more concise physical image, we choose to neglect the piezoelectricity when deriving the corresponding equations.

We apologize for the confusions and uncritical descriptions. We have reorganized the derivations with consideration the influence of piezoelectricity. Please kindly check it in the revised Supplementary Materials (part C of Note 2. Derivation of partial waves in ITBLN resonant cavity) in blue.

Q18. As shown in figure S3, in the coordinate angular bisectors systems, the relationship between the corresponding shear horizontal stress (T_5 , T_6) and T_z and T_y is not clearly.

Re: We apologize for the unclear description of the corresponding shear horizontal stress (T_5 , T_6) and (T_z , T_y). These components of stress are related in Fig. S5, in which the twisting angle (2θ) is set as 116° in (a) and 297° in (b), respectively. Taking 116° as an example, relationships between T_5 , T_6 and T_z , T_y can be explained as following steps.

First of all, according to Eq(R55), in a single layer of X-cut LiNbO₃, stress of the α and β polarized eigenmodes as well as their superposition can be decomposed into two components: T_5 along Z axis and T_6 along Y axis. Noted that T_1 along X axis remains zero for an SH polarized piezoelectrically stiffened mode.

As shown in Fig. R3(a), we construct the bilayers of LiNbO₃ from an original X-cut single layer of LiNbO₃. The upper layer (marked as “up” in the corresponding superscripts) experienced an inversion along z axis and then twisted with a rotation angle (2θ), while the lower layer (marked as “down” in the corresponding superscripts) remains unchanged as the original layer. Next, the original directions of α and β shear-horizontal polarization and their changes after the inversion (marked as “inversed” in the corresponding superscripts) and rotation (marked as “twisted” in the corresponding superscripts) was schematically shown in Fig. R3(b), as blue and red arrows, respectively. Meanwhile, axis of y and z can also be treated as directions of T_6 and T_5 . As shown in Fig. R3(c), the dashed arrows in light gray on the LiNbO₃ films indicate the y and z axis, which can also be marked as $T_6^{original}$ (or T_6^{down}) and $T_5^{original}$ (or T_5^{down}), respectively. After the inversion along z axis, the polarization and y axis of the layer was marked as $T_5^{inversed}$ (which is same as $T_5^{original}$) and $T_6^{inversed}$ (which is the opposite of $T_6^{original}$). Lastly, the dashed arrows in dark gray indicate the stress components $T_6^{twisted}$ (or T_6^{up}) along y axis and $T_5^{twisted}$ (or T_5^{up}) along z axis after the rotation from $T_6^{inversed}$ and $T_5^{inversed}$, respectively.

In Fig. S3, the coordinate bisectors system was defined as the angular bisectors between the polarizations of same mode in upper and lower layers. Here we provided a more detailed schematics. As shown in Fig. R4, the arrows indicating the z axis before and after the inversion and rotation and the corresponding polarizations of α and β modes in the rightmost of Fig. R3(b) was decomposed into two series: z axis and α polarization as well as z axis and β polarization. From both series, the angular bisector (marked as Z) of $z^{original}$ (or z^{down}) and $z^{twisted}$ (or z^{up}) is also be the angular

bisector of original α (or β) polarization and twisted α (or β) polarization. Therefore, the coordinate angular bisectors system can be simply obtained with the α and β polarizations in upper (after inversion and rotation) and lower layers, which denotes the Fig. R5, as part (b) of Fig. S3 in the revised Supplementary Materials.

Based on the discussion above, here we have defined the shear horizontal stress components T_Y and T_Z in the coordinate angular bisectors systems. As shown in Fig R6, the direction of T_Z is defined as the angular bisector between the $T_5^{inversed}$ (or T_5^{down}) and the $T_5^{twisted}$ (or T_5^{up}), as well as T_Y defined as the angular bisector between the $T_6^{inversed}$ (or T_6^{down}) and the $T_6^{twisted}$ (or T_6^{up}). We have also provided the schematic of the relationship between the corresponding shear horizontal stress (T_5, T_6) and (T_Z, T_Y) with the twisting angle (2θ) equaling to 297° , which is Fig. R7. To be noted that the defined bisectors systems T_Y and T_Z must exhibit the same chirality in the 116° rotation and 297° rotation, guarantying that Eq(S91) to Eq(S94) being validated. As a conclusion, Fig. R8 shows the schematics of coordinate angular bisector systems at a) 116° and b) 297° .

Figure R3. Schematic of (a) the original single layer of X-cut LiNbO₃, (b) the inversed layer of X-cut LiNbO₃ and (c) the layer of X-cut LiNbO₃ after the inversion and rotation, the dashed arrows in the bottom row indicate the corresponding shear horizontal stress.

Figure R4. Schematic of the definition of the shear horizontal stress components T_Y and T_Z in the coordinate angular bisectors systems at 116°

Figure R5. Schematic of the definition of the shear horizontal stress components T_Y and T_Z in the coordinate angular bisectors systems at 116°

Figure R6. Schematic of the definition of the shear horizontal stress components T_Y and T_Z in the coordinate angular bisectors systems at 116° .

Figure R7. Schematic of the definition of the shear horizontal stress components T_Y and T_Z in the coordinate angular bisectors systems at 297° .

Figure R8. schematics of coordinate angular bisector systems at a) 116° and b) 297° .

For ease of reading, we have reorganized the interpretation and figures (from Fig. S1 to Fig. S6) and inserted them in the main text of revised supplementary Materials, rather than setting in the last. *Moreover, we have added an additional section in blue as a complement to demonstrate the relationships between the coordinate angular bisectors systems and the corresponding shear horizontal stress (T_5 , T_6) and (T_Z , T_Y), as the reviewer concerns.*

Please kindly check it in the revised Supplementary Materials (Note 5. Details of the coordinate angular bisectors systems and corresponding shear horizontal stress).

Comments of the Reviewer #2

Two layers having equal thickness of LiNbO₃ are stacked together with twisted angle of X cut. Experimental as well as theoretical results are presented. It is observed that at angle of 111°, K value is maximum and there is no beta phase. Results are compared with other existing devices also.

Re: We thank the Reviewer #2 for the concise summary.

It is clearly showed that present results are very much superior to others. There are some points are observed.

Re: We would like to express our gratitude to Reviewer #2 for the highlighting the advancement of our work as “very much superior to others”. Below please find our point-by-point response to the points of the reviewer.

Questions of the Reviewer #2

Q1. The solution of equation 1 and 2 given in manuscripts is not clear. Is it done by analytical or FEM or software?

Re: We apologize for the unclear description of the solution of Eq(1) and (2) in manuscripts. The solutions are **analytical** and represents the three eigenmodes propagating along X axis in LiNbO₃ media. More precisely, they are solutions to the Eq(S14) and (S15) in the revised Supplementary Materials, given as:

$$\left\{ \frac{\partial^2}{\partial x^2} \begin{bmatrix} c_{11} & 0 & 0 \\ 0 & c_{66} & c_{56} \\ 0 & c_{56} & c_{55} \end{bmatrix} - \rho \frac{\partial^2}{\partial t^2} \mathbf{I}_{3 \times 3} \right\} \begin{bmatrix} u_x \\ u_y \\ u_z \end{bmatrix} = \frac{\partial^2}{\partial x \partial t} \begin{bmatrix} -e_{22}E_y + e_{31}E_z \\ e_{16}E_x \\ e_{15}E_x \end{bmatrix} \quad (\text{R35})$$

$$\left\{ \begin{bmatrix} 0 & 0 & 0 \\ 0 & \frac{\partial^2}{\partial x^2} & 0 \\ 0 & 0 & \frac{\partial^2}{\partial x^2} \end{bmatrix} - \mu_0 \frac{\partial^2}{\partial t^2} \begin{bmatrix} \epsilon_{11} & 0 & 0 \\ 0 & \epsilon_{11} & 0 \\ 0 & 0 & \epsilon_{33} \end{bmatrix} \right\} \begin{bmatrix} E_x \\ E_y \\ E_z \end{bmatrix} = \mu_0 \frac{\partial^2}{\partial x \partial t} \begin{bmatrix} e_{16}u_y + e_{15}u_z \\ -e_{22}u_x \\ e_{31}u_x \end{bmatrix} \quad (\text{R36})$$

Obviously, the longitudinal displacement component u_x couples with transverse electric field E_y and E_z , and decoupled with the other two displacement components u_y and u_z . Furthermore, with the relationship between the electric field and magnetic field:

$$\nabla \times \vec{E} = -\frac{\partial \vec{B}}{\partial t} \quad (\text{R3})$$

the polarization of adjoint magnetic field ($0, B_y, B_z$) would be perpendicular to the polarization of electric field ($0, E_y, E_z$). Therefore, this longitudinal piezoelectrically coupled mode (also called as quasi-longitudinal acoustic wave) can be treated as a coupling of a longitudinal acoustic wave and transverse electromagnetic wave (TEM). Based on the discussion above, here we plotted the schematic of polarization type of this longitudinal piezoelectrically coupled mode as Fig. R9, noting that the longitudinal displacement field u_x is marked as green arrows along x axis while transverse

magnetic field and electric field marked in light and dark gray, respectively.

Figure R9. Schematics of polarizations of electromagnetic and acoustic fields of quasi-longitudinal wave.

Figure R10. Schematics of polarizations of electric and acoustic fields of orthogonal shear-horizontal wave with a) α polarization and b) β polarization.

On the other hand, from Eq(R35) and (R36), despite the quasi-longitudinal solutions, there are two shear-horizontal polarized waves: shear-horizontal displacement components (u_y , u_z) and merely coupled longitudinal electric field E_x . The magnetic fields calculated by Eq(R3) comes to zero, and the polarization of displacement of the two modes are orthogonal (marked as α and β in manuscript). Based on this, here we plotted the schematics of their polarizations of acoustic and electric fields as Fig. R10. To be noted that the longitudinal electric field E_x was marked as dark gray arrows along x axis while the shear-horizontal displacement was marked in red for α polarization and blue for β polarization, respectively.

We have reorganized the part (a) of Fig.3 in the revised manuscript, and added the corresponding descriptions of solutions of Eq(1) and (2) in more details. Please kindly check them in the revised manuscript (line 121 to 134) in green.

Q2. Figure 6 is not explained clearly in supplementary file.

Re: We appreciate the reviewer's good suggestion. Fig.S6 is a summary of shear-horizontal acoustic modes in a single layer of X-cut LiNbO₃, as a comparison with the same thick ITBLN structure.

First of all, similar to the condition of ITBLN, the total distributions of acoustic and electric fields excited by an alternative longitudinal electric field E_x can be treated as superposition of eigen SH modes. Since there is no boundary in the middle thick of whole structure caused by the inversion and rotation, the resonant responses are distinguished respectively into two series: of α polarized SH standing waves and of β polarized SH standing waves. Each series would contain sequential orders of resonant modes, as shown in Fig. R11, which is the upper row of Fig. S6. To be noted that only odd orders with symmetric distributions of stress (solid lines in Fig. R11) can be excited, while the even orders with anti-symmetric distributions (dashed lines in Fig. R11) cannot be excited since their integral of U_m are zero.

Figure R11. Schematics of different orders of SH modes in single layer of X-cut LiNbO₃, the even orders with antisymmetric distribution of stress is plotted in dashed lines.

Next, the discussions above were verified by FEM simulations. Fig. R12 shows the simulated admittance curves of single layer of X-cut LiNbO₃ with thickness of 1000 um. There are four peaks in the frequency range of 1 to 8 MHz, relative to the first two orders of resonant modes of α and β polarizations.

Figure R12. Simulated admittance curve of single layer of X-cut LiNbO₃.

The mode shape from the vertical view of the four peaks in admittance curve are extracted and plotted as following Fig. R13. The white arrows are determined by the offset direction between the upper and lower surfaces, indicating the polarization type (α or β) of the resonant modes. Meanwhile, the transversal distributions of stress from the side view are mapped below their mode shapes. It is evident that all distributions of stress exhibit symmetric about the middle line of the layer.

Figure R13. Polarization types and stress distributions of first two order shear-horizontal modes, the white arrows indicate the polarization directions.

We have reorganized the corresponding descriptions and explanations in the revised Supplementary Materials. Please kindly check it in the revised Supplementary Materials (Note 3. Shear horizontal modes in single layer LN resonant cavity) in green.

Q3. Diameter of film is missing, only thickness is given.

Re: We apologize for not clearly describing the details of the experiment. As shown in the photo below, the diameter of the LiNbO₃ plate is 100 mm (4 inch), and the area of the Al electrodes on the surface is smaller than that of the ITBLN sample, where the largest electrode has a diameter of 50 mm.

We have added the description of diameter of film in the Methods of the revised manuscript. Please kindly check it (line 221) in green.

Figure R14 Photo of the ITBLN materials with double-sided aluminum electrodes.

Q4. Actual experiment set up photo or schematic is missing.

Re: We apologize for not clearly describing the details of the experiment. The schematic of the measurement system is illustrated in Fig. R15(a), which consists of three parts: the single/bi-layer piezoelectric sample with double-sided electrodes, printed circuit board (PCB) and the vector network analyzer (VNA). The PCB and VNA are connected through the RF coaxial cable, and the sample and the PCB are connected through the bonding wires. The actual experiment set up photo is presented in Fig. R15(b). The S_{11} response can be directly converted into admittance (Y), and shown in the screen of the VNA. The zoom-in photos [Fig.R15 (c-d)] show the details of the PCB and bonding wires in the front-side.

Figure R15 (a) Schematic of the measured system. (b) Photograph of the experimental setup and the zoom-in photographs of the (c) PCB and (d) bonding wires.

The description of the experiment set up has been added in green into the supplementary materials as a new section. Please kindly check them in the revised supplementary materials (Note 6. Experimental setup for measurement of ITBLN acoustic resonators) in green.

Q5. Figure S2 in supplementary, Is fig experimental or theoretical?

Re: Thanks to the reviewer. Figure S2 is obtained by FEM simulations.

Q6. For each thickness, there may be different angle of optimization. At present case, for equal thickness, it is 111. Can it be verified or justified?

Re: We gratefully appreciate the reviewer for the valuable suggestion. Reviewer is right. For different thickness ratio, the values of optimal angles are different. Particularly, when thickness of each layer equals to each other, the optimal angle is 111° in experiment.

Figure R16. Simulated admittance curves of ITBLN with different angles under different thickness ratios.

The influences of absolute thickness and ratio between two thickness of upper and lower layers need to be considered separately. First of all, changing the thickness ratio would cause the variations of both the resonant frequency and effective electromechanical coupling coefficient, as Fig. 2 shows in the manuscripts. For each thickness ratio, there are two optimal angles, as one for maximum effective electromechanical coupling coefficient of mode I and the other for mode II. However, we recognized the optimal angles as the “magic angle” only if the two thickness are equal (thickness ratio is 0.5), because under this condition the excitation of the other mode comes to zero while one of modes exhibits largest effective electromechanical coupling coefficient. Fig. R16 shows the simulated admittance curves of mode I and II with different angles under different thickness ratios, it can be confirmed for each thickness ratio, there exists two optimal angles, but only when the thickness ratio is 0.5, the corresponding optimal angles becomes the magic angle.

On the other hand, if the thickness ratio is set as 0.5, but the total thickness changes, the characteristics of excitation strength (effective electromechanical coupling coefficient) across rotation angles would not be affected, and only the resonant frequency changes. As Fig. R16 shows, although the absolute values of resonant frequency of mode I and II changed with total thickness, the magic angles still exist and equals to about (117°, 297°) in simulations.

Q7. The formulation gives in supplementary more useful to understand the paper. If possible, give some of relevant equations in manuscripts or at least cite about supplementary information in the manuscripts for detailed derivation.

Re: We gratefully appreciate the reviewer for the valuable suggestion. We have reviewed all derivations and formulas presented in both the manuscript and the Supplementary Materials.

Some key formulas, presented in the supplementary as the dispersion equations Eq(S17), are added into the section of “Theoretical analysis of piezoelectrically coupled waves in twisted bilayer lithium niobate structure” in the manuscripts for a more clear descriptions of solutions to Eq(1) and (2). Relationships from Eq(S91) to Eq(S94) between the symmetric or anti-symmetric distributions of stress along the angular bisector systems (T_Z , T_Y) and the corresponding shear horizontal stress (T_5 , T_6) were added into the section of “Determination of optimal angle for electromechanical coupling coefficient” in manuscripts, since those relationships are rather important to derive the optimal angles. Other equations in supplementary materials for more details have been cited with careful attentions.

As a conclusion, here we have reorganized in green the citations of formulations in Supplementary Materials in our revised manuscript and listed as follows:

Section number of supplementary materials	Citation positions in manuscript
Note 1. Derivation of piezoelectric coupled acoustic waves in X-cut LiNbO3 crystal	line 114&115
	line 122
	line 133
Note 2. Derivation of partial waves in ITBLN resonant cavity	line 149&150
	line 156&157
	line 153&154
	line 187&188

Please kindly check them in the revised manuscript.

Comments of the Reviewer #3

This work constructs a bilayer LiNbO₃ with inverted polarization and twisted angle, and shows an effective electromechanical coupling coefficient of 85.5%.

Re: We thank the Reviewer #3 for the concise summary.

The work is essentially a classical wave propagation problem in a layered structure, and the phenomenon is well understood from superposition point of view, as authors articulated in the manuscript.

Re: We appreciate the reviewer for the approval of our efforts to understand the exotic characteristics of piezoelectric resonant induced by bonding two layers of X-cut LiNbO₃ together with a twisting angle. As reviewer's comments, the carrier of our research is classical elastic wave, which is excited by piezoelectricity. However, compared to the general propagation and resonance problems, the phenomenon reported in our manuscript, i.e., huge effective electromechanical coupling coefficient for one mode while the other mode disappears, is novel. This phenomenon originates from the symmetry properties of our proposed ITBLN structure, which is beyond the scope of discussions on the classical wave propagation problems. In fact, it is exactly the classical background as a major barrier to understand the essential of this phenomenon, since the whole system exhibits the coupling of electric and elastic fields while the media with piezoelectricity is always anisotropic.

In our manuscript, we aim to provide an intuitive and clear physical image for this symmetry, and associate this symmetry properties with the calculation of effective electromechanical coupling coefficients of resonant modes. Based on this, we choose the Berlincourt equations (equation 4 in the revised manuscript) to relate the effective electromechanical coupling coefficients with the distributions of elastic and electric fields of resonant modes. Thus, view of superposition of bulk waves satisfying the field equations (equation 1 and 2 in the revised manuscript) is suitable for elaborating this symmetry properties. In addition, the accurate field distributions are calculated by FEM method (which is, in our opinion, essentially containing the classical wave propagation problem), but results given by FEM simulations can hardly derive and explain the magic angle phenomenon in a straightforward and concise view. Therefore, both the superposition and FEM are tools for understanding what happens for an acoustic resonator in a twisted bilayer structure. More specially speaking, FEM simulations for full distributions of fields provided a direct guide for experiments and analysis while the superposition understanding demonstrated the conceptual discoveries.

I thus fail to see the conceptual advances that were demonstrated in true twistrionics.

Re: We express our gratitude to Reviewer for pointing this out. Our research focuses on the exotic characteristics of piezoelectrically coupled elastic waves in proposed ITBLN structure, which cannot be recognized as "true twistrionics". However, we think our work exhibits the following conceptual advances:

Firstly, it is the first time that the piezoelectrical magic angles for a twisted bilayer LiNbO₃ structure was reported, and its mechanism was analyzed theoretically. As in our response to reviewer's comments before, the exotic phenomenon, manifested as huge effective

electromechanical coupling coefficients for one mode while the other mode disappears, is essentially a result of symmetry properties induced by bonding a pair of uniform-thick X-cut LiNbO₃ layers after inversion and rotation. From this view, the classical background of our research does not dismiss the conception. On the contrary, according to the best of our known, introduction of this kind of twisting-induced symmetry properties into an acoustic resonator at megahertz scale is first proposed and analyzed in our research.

Secondly, the preparation of twisted bilayer LiNbO₃ in our manuscripts is realized by direct bonding technique, which guarantees the quality of the interface between the two layers. This is much important for an acoustic system, because only the interface satisfying the continuous boundary conditions enables the further analysis, utilizations and designs in devices. With validation of the preparation of twisted bilayer crystal materials in our work, a new degree characterized by twisted bonding is induced conceptually and realized technically. Therefore, we hope to provide a new choose on the modulation of piezoelectricity and designs of acoustic systems.

Thirdly, we tend to conclude the advances of our work from view of applications. As is well known, acoustic resonators, as components of filters, have been widely studied and used in 5th generation (5G) and future 6G wireless communication technology. The effective electromechanical coupling coefficient is one of most important indexes of an acoustic resonator since its value determines the width of passband of filter. Moreover, in conventional design of acoustic resonators, the effective electromechanical coupling coefficient is usually adjusted by changing the in-plane orientation of propagation of waves, and it is difficult to simultaneously achieve the largest coupling and a clean band without other spurious modes. This critical issue does not appear in our proposed twisting bilayer LiNbO₃ resonators. We believe that twisting a pair of piezoelectric layers together provides a novel way to modulate the piezoelectricity, particularly the coupling coefficient.

The manuscript claims giant piezoelectricity and modulation of polarization. Neither of these claims are demonstrated at the material level, and the effective electromechanical coupling coefficient is a result of superposition of waves in two layers, which cancels spurious peaks in the single layer, I believe.

Re: As our responses above, we do agree that our work bases on classic elastic waves. The intrinsic properties of piezoelectricity in each layer of LiNbO₃ are not changed, different from the true twistrionics scenario. However, we would like to emphasize the significance of twisting giant piezoelectricity and modulation of polarization. Specifically, either piezoelectricity and modulation of polarization in our manuscript is demonstrated as fields overlapping, which is classical but exhibits unique symmetry properties. We hope the reviewer can reconsidered that it is reasonable for the level of scale (not at the material level but in macro material properties) we demonstrated our findings suiting the classical theory of wave equations. In addition, **a major challenge when we demonstrated our findings is the validation of the continuous boundary conditions, which relates the partial waves and their distributions of fields in two layers.** Fig. R1 shows the transmission electron microscope (TEM) characterization image of the bi-layer LiNbO₃ structure, in which almost no intermediate or buffer layer can be observed at the interface. We conclude that this near-ideal interface between two hetero layers (in our works “hetero” means mismatch of

crystallographic axis induced by inversion and rotation) is guaranteed by advancement of direct bonding technique, which also enable the preparation of twisted bilayer crystal materials.

Figure R1 (a) TEM image of the interface of bilayer LiNbO₃ plate. (b) Nb and (c) O element distribution at the interface region.

The resulting 85.5% coupling coefficient itself is comparable to the corresponding value in single layer as well as state of art devices, not that remarkable as claimed.

Re: We appreciate the reviewer's good suggestion. The result was based on the innovation of twisted piezoelectricity, and the remaining relevant parameters were unoptimized. We compared this huge coupling coefficient in our manuscript not only to show novelty of the magic angle phenomenon directly but also to demonstrate the potential of concepts of twisted piezoelectricity in applications.

Having addressed all the concerns raised, we hope the reviewer will reconsider his/her initial perspective.

REVIEWERS' COMMENTS

Reviewer #1 (Remarks to the Author):

All my comments are well addressed; thus, I recommend this paper to be published.

Reviewer #3 (Remarks to the Author):

I have examined the Response of the authors, and I remain neutral in the recommendation,